# Opioid suppression of an excitatory pontomedullary respiratory circuit by convergent mechanisms

Jordan T Bateman, Erica S Levitt*[†]

Department of Pharmacology and Therapeutics, Breathing Research and Therapeutics Center, University of Florida, Gainesville, United States

**Abstract** Opioids depress breathing by inhibition of interconnected respiratory nuclei in the pons and medulla. Mu opioid receptor (MOR) agonists directly hyperpolarize a population of neurons in the dorsolateral pons, particularly the Kölliker-Fuse (KF) nucleus, that are key mediators of opioid-induced respiratory depression. However, the projection target and synaptic connections of MOR-expressing KF neurons are unknown. Here, we used retrograde labeling and brain slice electrophysiology to determine that MOR-expressing KF neurons project to respiratory nuclei in the ventrolateral medulla, including the preBötzinger complex (preBötC) and rostral ventral respiratory group (rVRG). These medullary-projecting, MOR-expressing dorsolateral pontine neurons express FoxP2 and are distinct from calcitonin gene-related peptide-expressing lateral parabrachial neurons. Furthermore, dorsolateral pontine neurons release glutamate onto excitatory preBötC and rVRG neurons via monosynaptic projections, which is inhibited by presynaptic opioid receptors. Surprisingly, the majority of excitatory preBötC and rVRG neurons receiving MOR-sensitive glutamatergic synaptic input from the dorsolateral pons are themselves hyperpolarized by opioids, suggesting a selective opioid-sensitive circuit from the KF to the ventrolateral medulla. Opioids inhibit this excitatory pontomedullary respiratory circuit by three distinct mechanisms—somatodendritic MORs on dorsolateral pontine and ventrolateral medullary neurons and presynaptic MORs on dorsolateral pontine neuron terminals in the ventrolateral medulla—all of which could contribute to opioid-induced respiratory depression.

*For correspondence:
elsawyer@umich.edu

Present address: [†]Department of Pharmacology, University of Michigan, Ann Arbor, United States

Competing interest: The authors declare that no competing interests exist.

## Editor's evaluation

Opioid-induced respiratory depression is one of the side effects of opioid drugs. Although opioid overdose deaths are highly prevalent, our knowledge of the neural circuits underlying respiratory depression in the brainstem is far from complete. The present study used a variety of sophisticated experimental techniques to convincingly reveal the identity of brainstem components that are part of the neural circuits involved in the mediation of opioid respiratory effects, together with defining potential synaptic underlying mechanisms. They focused on two regions of the brainstem, namely the Kolliker-Fuse and the preBötzinger Complex, and proposed a combination of three complementary processes at pre- and post-synaptic sites in both KF and preBötC regions to explain respiratory depression linked to opioid exposure. This study provides very important findings on the circuitry involved in opioid-induced respiratory depression, and the present results are of broad interest to the respiratory control research community, as well as medically relevant.

## Introduction

With the prevalence of opioid overdose on the rise (*Wilson et al., 2020*; *Mattson et al., 2021*), understanding the network mechanisms of opioid-induced respiratory depression is of particular importance. Opioids, due to activation of the mu opioid receptor (MOR) (*Dahan et al., 2001*), depress breathing by inhibiting interconnected respiratory nuclei in the pons and medulla (*Bateman et al., 2021*; *Ramirez et al., 2021*). Despite significant progress, detailed mechanisms by which this occurs remain elusive, especially for the dorsolateral pons. We sought to identify mechanistic insight concerning how opioids inhibit pontomedullary respiratory neurocircuitry that gives rise to opioid-induced respiratory depression.

Respiration is generated and controlled by an interconnected pontomedullary network in the brainstem (*Del Negro et al., 2018*). The Kölliker-Fuse (KF) nucleus and adjacent lateral parabrachial area (LPB) of the dorsolateral pons are critical to the neural control of breathing (*Lumsden, 1923*; *Fung and St John, 1995*; *Dutschmann and Herbert, 2006*; *Smith et al., 2007*). The KF/LPB is composed of a heterogeneous population of respiratory neurons that impact respiratory rate and pattern (*Chamberlin and Saper, 1994*; *Navarrete-Opazo et al., 2020*; *Saunders and Levitt, 2020*) via excitatory projections to respiratory nuclei in the ventrolateral medulla, including, but not limited to the Bötzinger complex (BötC), preBötzinger complex (preBötC), and rostral ventral respiratory group (rVRG) (*Song et al., 2012*; *Yokota et al., 2015*; *Geerling et al., 2017*; *Yang et al., 2020*). The preBötC generates inspiratory rhythm (*Smith et al., 1991*), which is relayed to inspiratory premotor neurons in the rVRG. The BötC contains mostly inhibitory neurons that fire during expiration and is a major source of inhibition within the network (*Schreihofer et al., 1999*; *Ezure et al., 2003*). The dynamic interplay between the KF/LPB and the BötC, preBötC, and rVRG is essential for optimized respiratory output (*Dutschmann and Dick, 2012*; *Smith et al., 2007*). Unfortunately, all of these respiratory nuclei express MORs, leading to inhibition of the control of breathing network via multiple potential sites and mechanisms (*Gray et al., 1999*; *Lonergan et al., 2003*; *Montandon et al., 2011*; *Levitt et al., 2015*; *Cinelli et al., 2020*).

Two respiratory nuclei considered critical for opioid-induced respiratory depression are the KF/LPB of the dorsolateral pons and the preBötC of the ventrolateral medulla (*Bachmutsky et al., 2020*; *Varga et al., 2020*). The dorsolateral pontine KF/LPB is considered a key contributor of opioid-induced respiratory depression because (1) deletion of MORs from the KF/LPB attenuates morphine-induced respiratory depression (*Bachmutsky et al., 2020*; *Varga et al., 2020*; *Liu et al., 2021*), (2) opioids injected into the KF/LPB reduce respiratory rate (*Prkic et al., 2012*; *Levitt et al., 2015*; *Miller et al., 2017*; *Liu et al., 2021*), (3) blockade of KF/LPB opioid receptors rescues fentanyl-induced apnea (*Saunders and Levitt, 2020*), and (4) chemogenetic inhibition of MOR-expressing LPB neurons induces respiratory depression (*Liu et al., 2021*). Yet, mechanisms by which the opioid inhibition of dorsolateral pontine neurons alter neurotransmission in the respiratory circuitry and causes suppression of breathing are unknown.

MORs inhibit neurotransmission by hyperpolarizing neurons through activation of somatodendritic GIRK channels and/or inhibiting presynaptic neurotransmitter release through inhibition of voltage-gated calcium channels (*Jiang and North, 1992*; *Chahl, 1996*; *Zamponi and Snutch, 1998*; *Al-Hasani and Bruchas, 2011*). In the preBötC, presynaptic MORs inhibit synaptic transmission (*Ballanyi et al., 2010*; *Wei and Ramirez, 2019*; *Baertsch et al., 2021*) and are expressed more abundantly than somatodendritic MORs (*Lonergan et al., 2003*). These presynaptic MORs in the preBötC are poised to play a major role in the mechanism of opioid suppression of breathing within the inspiratory rhythm-generating area, but the specific origins of MOR-expressing synaptic projections remain unknown. Here, we tested the hypothesis that they are coming from the dorsolateral pons.

Opioids hyperpolarize a subset of KF neurons (*Levitt et al., 2015*), whose neurochemical identity and possible projection targets are unknown. Glutamatergic KF neurons project to the ventrolateral medulla (*Song et al., 2012*; *Yokota et al., 2015*; *Geerling et al., 2017*) and, if inhibited by opioids—either by somatodendritic activation of GIRK channels and/or presynaptic inhibition of neurotransmitter release—could depress breathing. Therefore, we hypothesized that MOR-expressing KF neurons project to and form excitatory synapses onto respiratory controlling neurons in the ventrolateral medulla (i.e. the preBötC and rVRG), and that this excitatory synapse is inhibited by presynaptic MORs on KF terminals. The results show that this excitatory pontomedullary respiratory circuit is robustly inhibited by opioids by three different mechanisms, involving presynaptic and postsynaptic

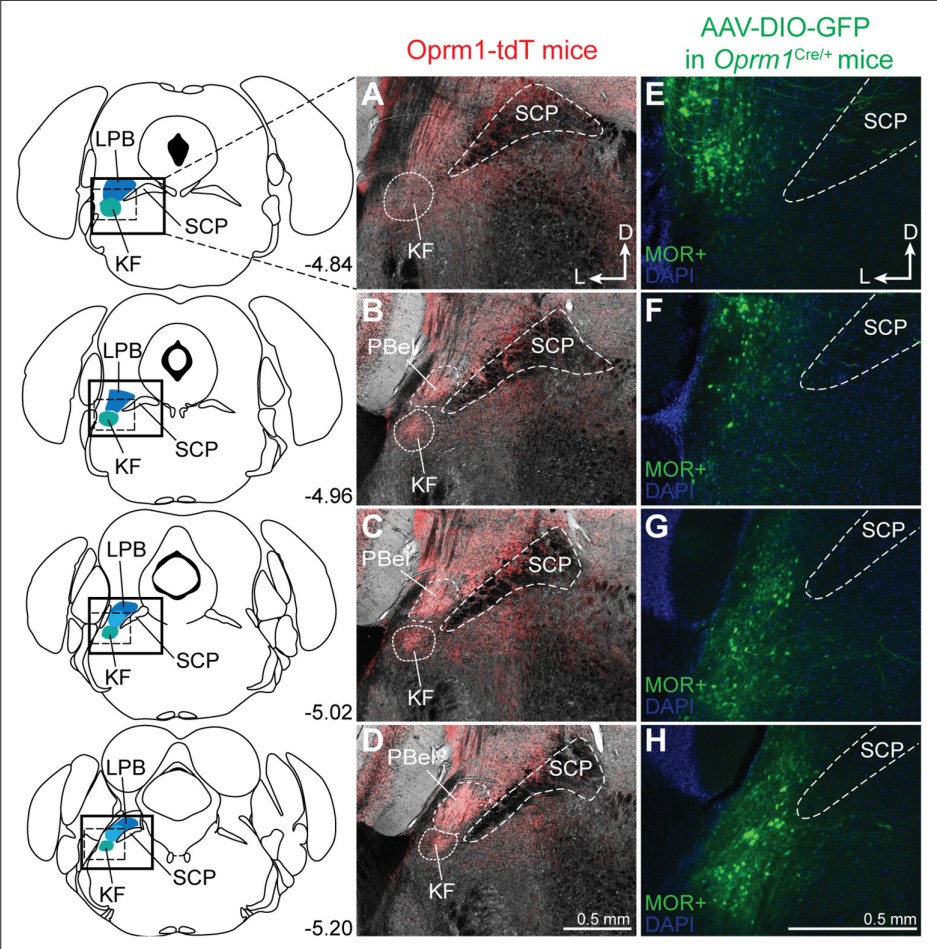

**Figure 1.** Dorsolateral pontine neurons express mu opioid receptors (MORs). (**A–D**) Representative images of tdTomato expression, as an indicator of MOR expression, in coronal dorsolateral pontine slices from Oprm1-tdT mice (n = 3) across the rostral to caudal Kölliker-Fuse/lateral parabrachial area (KF/LPB) axis. Fluorescent tdTomato image is overlaid onto brightfield image to show landmarks. (**E–H**) Representative images of GFP expression, as an indicator of MOR expression, following injection of virus encoding Cre-dependent GFP into KF/LPB to label MOR+ neurons in adult *Oprm1*[Cre/+] mice (n = 5). Left column are slice schematics corresponding to each row. The approximate levels caudal to bregma (in mm) are to the right of each schematic. The images correspond to the solid boxed area (**A–D**) or the dotted boxed area (**E–H**) of the slice schematic. The scale bar in (**D**) applies to images (**A–D**). The scale bar in (**H**) applies to images (**E–H**). PBel, external lateral subdivision of parabrachial; SCP, superior cerebellar peduncle.

opioid receptors in the dorsolateral pons and the ventrolateral medulla, revealing convergent mechanisms by which opioids can depress breathing.

## Results

### Oprm1 expression in dorsolateral pontine neurons

To visualize MOR-expressing dorsolateral pontine neurons, *Oprm1*[Cre/Cre] mice (*Baertsch et al., 2021*; *Liu et al., 2021*) were crossed with tdTomato Cre-reporter mice to generate *Rosa26*[LSL-tdT/+]::*Oprm1*[Cre/+] mice (hereby referred to as Oprm1-tdT mice) that express tdTomato in neurons that also express MORs at any point during development. MOR-expressing neurons and neurites were identified in the dorsolateral pons, specifically in the lateral parabrachial area and KF (n = 3; *Figure 1A–D*).

To selectively label neurons that express MORs during adulthood, a virus encoding Cre-dependent GFP expression (AAV-DIO-GFP) was injected into the dorsolateral pons of *Oprm1*[Cre/+] 2–4-month-old mice (n = 5). MOR-expressing neurons were again identified in the lateral parabrachial and KF areas

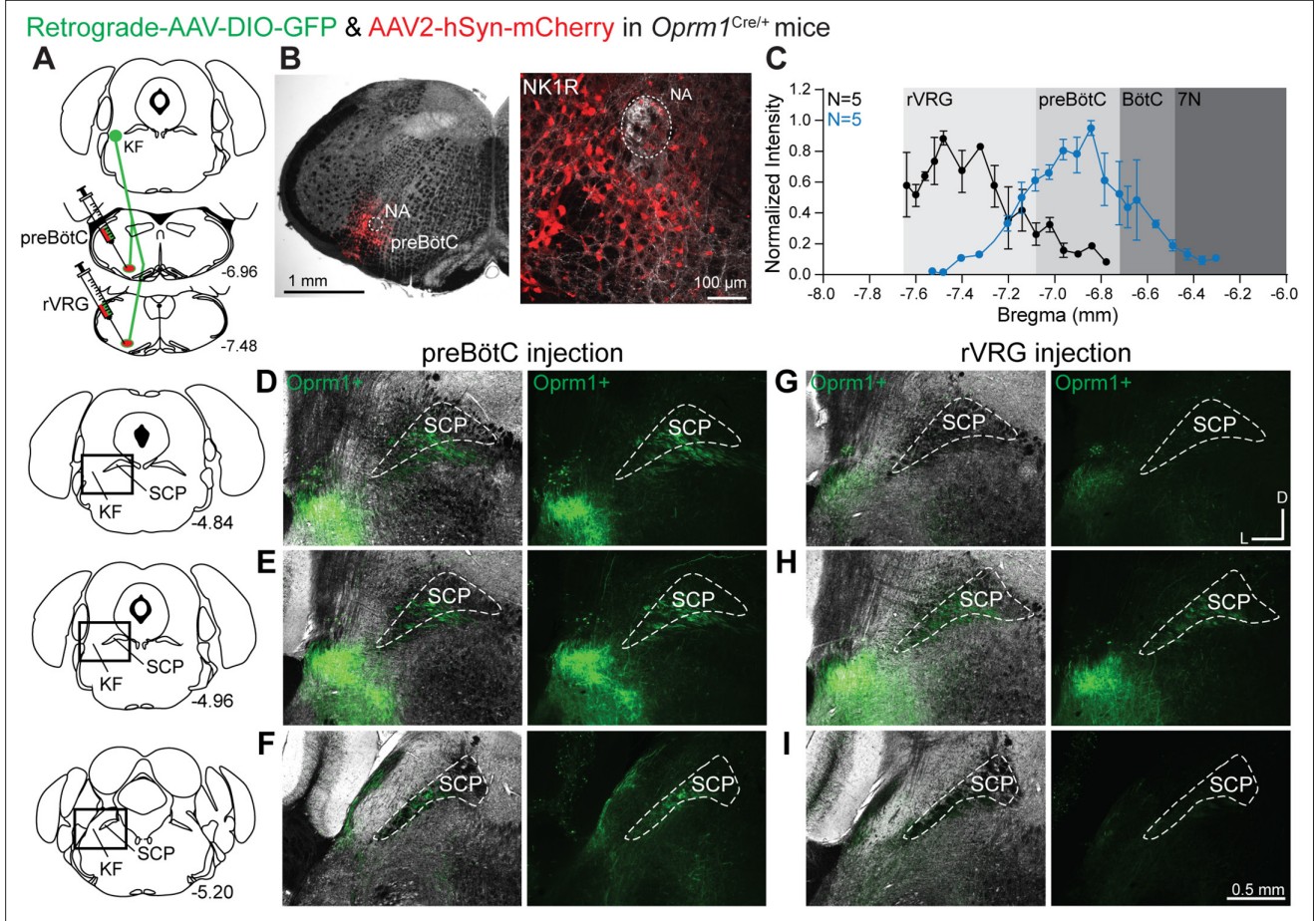

**Figure 2.** *Oprm1*+ Kölliker-Fuse (KF) neurons and neurites retrogradely labeled from the preBötzinger complex (preBötC) and rostral ventral respiratory group (rVRG). (**A**) Schematic illustrating the approach to retrogradely label *Oprm1*+ KF neurons and neurites projecting to the preBötC or rVRG. (**B**) Images of coronal slices from the medulla with a control injection of AAV2-hSyn-mCherry into the preBötC of an *Oprm1*^Cre/+ mouse to mark the injection site. Immunohistochemistry for the neurokinin 1 receptor (NK1R) was used as a marker for the preBötC and the nucleus ambiguous (NA). (**C**) Quantification of normalized AAV2-hSyn-mCherry fluorescence intensity along the rostral to caudal axis in the ventrolateral medulla of preBötC (n = 5) and rVRG (n = 5). Anatomical level relative to Bregma is indicated on the x-axis. (**D–I**) Representative images of GFP expression, as an indicator of retrograde-labeled *Oprm1*-expressing neurons and neurites, following injections into the preBötC (**D–F**) or the rVRG (**G–I**) across three levels of the dorsolateral pons. The bregma level is indicated on the schematics to the left of each row. The scale bar in (**I**) applies to all images (**D–I**). Higher magnification images of bregma –4.84 are shown in *Figure 2—figure supplement 2*.

The online version of this article includes the following source data and figure supplement(s) for figure 2:

**Source data 1.** Quantification of spread at the injection sites.

**Figure supplement 1.** *Oprm1*+ Kölliker-Fuse (KF) neurons project to the Bötzinger complex (BötC).

**Figure supplement 2.** Higher magnification images of retrogradely labeled Kölliker-Fuse (KF) neurons.

(*Figure 1E–H*). Neuronal cell bodies were more apparent in these images since MOR-expressing afferents into the dorsolateral pons were not labeled by this approach. These results are consistent with previous studies showing that MORs are expressed in LPB (*Huang et al., 2021*; *Liu et al., 2021*) and KF (*Levitt et al., 2015*; *Varga et al., 2020*).

## *Oprm1*+ KF neurons project to respiratory nuclei in the ventrolateral medulla

We hypothesized that *Oprm1*+ KF neurons project to respiratory controlling nuclei in the ventrolateral medulla, especially the preBötC and rVRG. To determine this, retrograde virus encoding Cre-dependent expression of GFP (retrograde AAV-hSyn-DIO-eGFP) was unilaterally injected into the preBötC or the rVRG of *Oprm1*^Cre/+ mice (*Figure 2*). As a control, anterograde virus encoding mCherry

(AAV2-hSyn-mCherry) was co-injected to mark the injection site. The intensity of mCherry expression was measured throughout the rostral-caudal axis of the ventrolateral medulla to quantify the extent of injection spread in accordance with medullary anatomical markers (*Figure 2B and C*). In addition, immunolabeling for the neurokinin 1 receptor (NK1R) was used as a marker of the preBötC (*Gray et al., 1999*; *Montandon et al., 2011*) and to identify the nucleus ambiguous (NA), which was especially useful for the compact section of the NA in the preBötC region (*Figure 2B*). Injection sites were categorized based on the location of peak mCherry expression intensity (*Figure 2C* and *Figure 2— figure supplement 1*).

*Oprm1*+ dorsolateral pontine neurons and neurites were retrogradely labeled from both preBötC and rVRG (*Figure 2D–I*). Interestingly, *Oprm1*+ projections to the preBötC (*Figure 2D–F*; n = 5) and the rVRG (*Figure 2G–I*; n = 5) were mostly localized to the rostral and mid-rostral KF, and nearly absent in the caudal KF and lateral parabrachial area (*Figure 2F and I*). The majority of the retrogradely labeled dorsolateral pontine neurons and neurites were ipsilateral to the injection site, with very few or no contralateral neurons or neurites expressing GFP. Injections in three mice were located rostrally from the preBötC with the peak of mCherry expression in the BötC (*Figure 2—figure supplement 1*). In contrast to preBötC and rVRG projections, qualitatively fewer *Oprm1*+ KF neurons projected to the BötC (*Figure 2—figure supplement 1E–G*). Higher magnification images of retrograde-labeled GFP-expressing *Oprm1*+ KF neurons are shown in *Figure 2—figure supplement 2*.

## Presynaptic opioid receptors inhibit glutamate release from KF terminals onto excitatory medullary neurons

Given that KF neurons projecting to the ventrolateral medulla are glutamatergic (*Geerling et al., 2017*) and express MORs (*Figure 2*), we hypothesized that opioids inhibit glutamate release from KF terminals onto respiratory neurons in the ventrolateral medulla, particularly the preBötC and rVRG. To test this hypothesis, we unilaterally injected a virus encoding channelrhodopsin2 (AAV2-hSyn-hChR2(H134R)-EYFP-WPRE-PA) into the KF of vglut2$^{Cre/LSL-tdT}$ mice (*Figure 3A and B*). We made whole-cell voltage-clamp recordings from tdTomato-expressing, excitatory vglut2-expressing preBötC and rVRG neurons contained in acute brain slices (*Figure 3C*). Because we could not determine the respiratory-related firing pattern of the neurons we recorded from in this study, we chose to target vglut2-expressing neurons since (1) this contains the population of inspiratory rhythm-generating preBötC neurons (*Wallén-Mackenzie et al., 2006*; *Gray et al., 2010*; *Cui et al., 2016*) and inspiratory premotor rVRG neurons, (2) KF neurons project to excitatory, more so than inhibitory, preBötC neurons (*Yang et al., 2020*), and (3) deletion of MORs from vglut2 neurons eliminates opioid-induced depression of respiratory output in medullary slices (*Sun et al., 2019*; *Bachmutsky et al., 2020*). Optogenetic stimulation of KF terminals drove pharmacologically isolated excitatory postsynaptic currents (oEPSCs) in excitatory preBötC and rVRG neurons (*Figure 3D and I*) that were blocked by the AMPA-type glutamate receptor antagonist 6,7-dinitroquinoxaline-2,3-dione (DNQX; 10 μM; *Figure 3—figure supplement 1A and B*, n = 11). Additionally, KF synapses onto medullary respiratory neurons are monosynaptic because oEPSCs were eliminated by tetrodotoxin (TTX; 1 μM) yet restored by subsequent application of 4-aminopyridine (4AP; 100 μM) (*Figure 3—figure supplement 1A and C*; n = 7). Thus, KF neurons send monosynaptic, glutamatergic projections to excitatory ventrolateral medullary neurons.

To determine whether opioids inhibit glutamate release from KF terminals onto medullary respiratory neurons, pairs of oEPSCs (50 ms inter-stimulus interval) were recorded from excitatory preBötC and rVRG neurons, and the endogenous opioid agonist [Met$^5$]enkephalin (ME) was applied to the perfusion solution. ME (3 μM) decreased the oEPSC amplitude in preBötC neurons (*Figure 3D and E*; n = 13) and in rVRG neurons (*Figure 3I and J*; n = 9), which reversed when ME was washed from the slice. In addition, ME increased the paired-pulse ratio (PPR) in both preBötC (*Figure 3F*; n = 11) and rVRG neurons (*Figure 3K*; n = 9), indicating inhibition of glutamate release by presynaptic MORs. The proportion of opioid-sensitive KF terminals was surprisingly high, considering that oEPSCs were inhibited by ME by a threshold of at least 30% in nearly all preBötC neurons (11 of 13 neurons) and all rVRG neurons. Thus, presynaptic opioid receptors inhibit glutamate release from KF terminals onto a majority of excitatory preBötC and rVRG neurons (91% [20 of 22 neurons]).

We were also able to determine whether the excitatory preBötC or rVRG neuron that received opioid-sensitive glutamatergic synaptic input from the KF was itself hyperpolarized by opioids by monitoring the holding current. ME (3 μM) induced an outward current in 68% of preBötC neurons

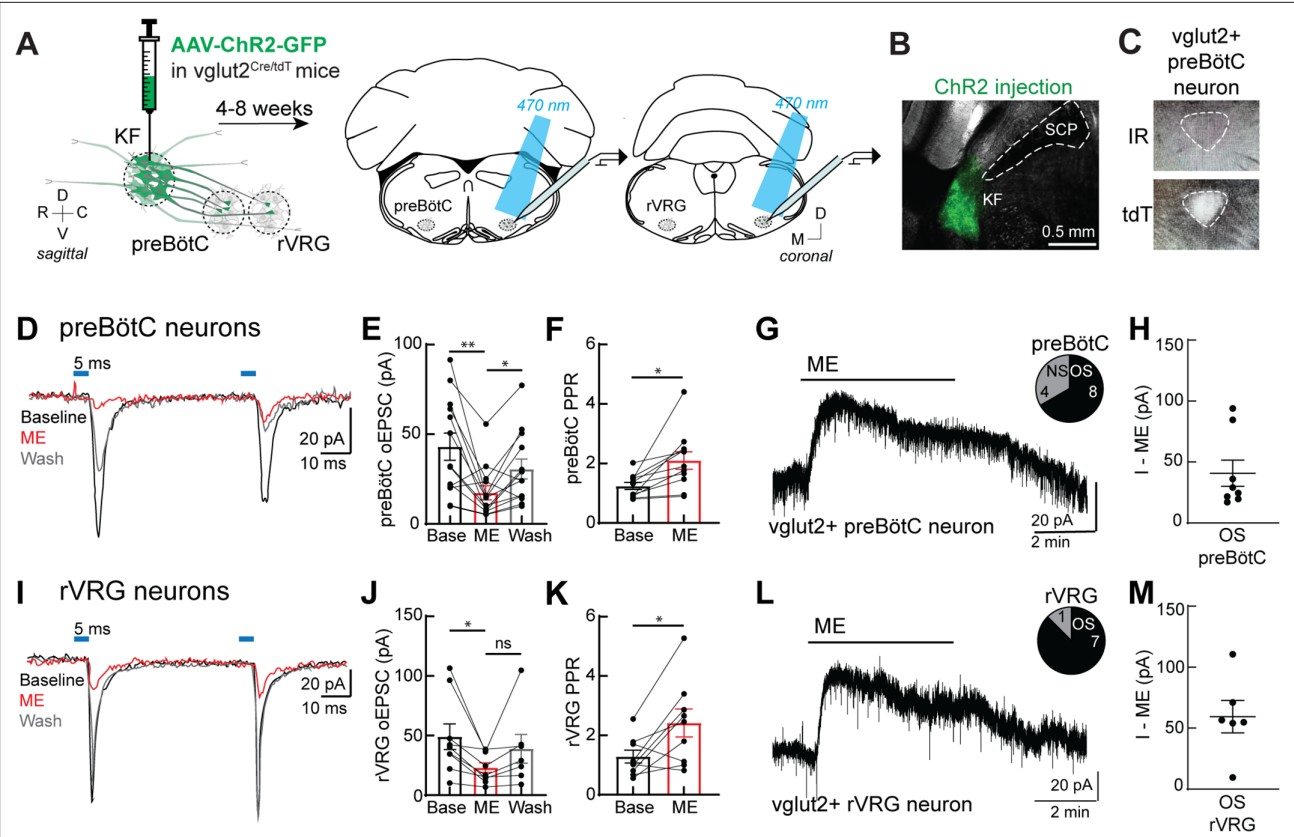

**Figure 3.** Presynaptic opioid receptors inhibit glutamate release from Kölliker-Fuse (KF) terminals onto excitatory preBötzinger complex (preBötC) and rostral ventral respiratory group (rVRG) neurons. (**A**) Schematic of approach to optogenetically stimulate KF terminals and drive optogenetically evoke excitatory postsynaptic currents (oEPSCs) in excitatory preBötC and rVRG neurons in an acute brain slice. (**B**) Representative image of ChR2-GFP expression in the KF (injection area) of vglut2-tdT mouse. (**C**) tdTomato-expressing, excitatory vglut2-expressing preBötC (or rVRG) neurons were identified in acute brain slices. (**D**) Recording of pairs of oEPSCs (5 ms stimulation, 50 ms inter-stimulus interval) from an excitatory preBötC neuron in an acute brain slice at baseline (black), during perfusion of Met-enkephalin (ME, 3 µM) (red), and after wash (gray). (**E**) ME decreased oEPSC amplitude in preBötC neurons (n = 13; **p=0.007, *p=0.013 by one-way ANOVA and Tukey's post-test). (**F**) ME increased the paired-pulse ratio (P2/P1) in preBötC neurons (n = 11; *p=0.001 paired *t*-test). (**G**) ME (3 µM) induced outward currents in 8 of 12 preBötC neurons. OS, opioid-sensitive; NS, non-opioid-sensitive. (**H**) The amplitude of the outward current (I–ME, pA) in OS preBötC neurons. (**I**) Recording of pairs of oEPSCs (5 ms stimulation, 50 ms inter-stimulus interval) from an excitatory rVRG neuron in an acute brain slice at baseline (black), during perfusion of ME (3 µM) (red), and after wash (gray). (**J**) ME decreased oEPSC amplitude in rVRG neurons (n = 9; *p=0.027 by one-way ANOVA and Tukey's post-test). (**K**) ME increased the paired-pulse ratio (P2/P1) in rVRG neurons (n = 9; *p=0.043 by paired *t*-test). (**L**) ME-mediated outward currents were observed in 7 of 8 rVRG neurons. (**M**) The amplitude of the outward current (I–ME, pA) in OS rVRG neurons. For all graphs, bar/line and error represent mean ± SEM. Individual data points are from individual neurons.

The online version of this article includes the following source data and figure supplement(s) for figure 3:

**Source data 1.** Presynaptic opioid receptors inhibit glutamate release from Kölliker-Fuse (KF) terminals onto excitatory preBötzinger complex (preBötC) and rostral ventral respiratory group (rVRG) neurons.

**Figure supplement 1.** Kölliker-Fuse (KF) neurons send monosynaptic, glutamatergic projections to excitatory ventrolateral medullary neurons.

(8 of 12 neurons) (*Figure 3G and H*) and 88% of rVRG neurons (7 of 8 neurons) (*Figure 3L and M*). There was no difference in the amplitude of the ME-mediated current in preBötC and rVRG neurons (p=0.294; unpaired *t*-test). Thus, a majority of excitatory preBötC and rVRG neurons that receive opioid-sensitive glutamatergic synapses from KF neurons are themselves hyperpolarized by opioids, indicating both pre- and postsynaptic suppression of this excitatory synapse by opioids.

## Opioids hyperpolarize medullary-projecting KF neurons

Opioids hyperpolarize a subpopulation (~60%) of KF neurons by activating G protein-coupled inwardly rectifying potassium (GIRK) channels (*Levitt et al., 2015*). Given that KF neurons that project

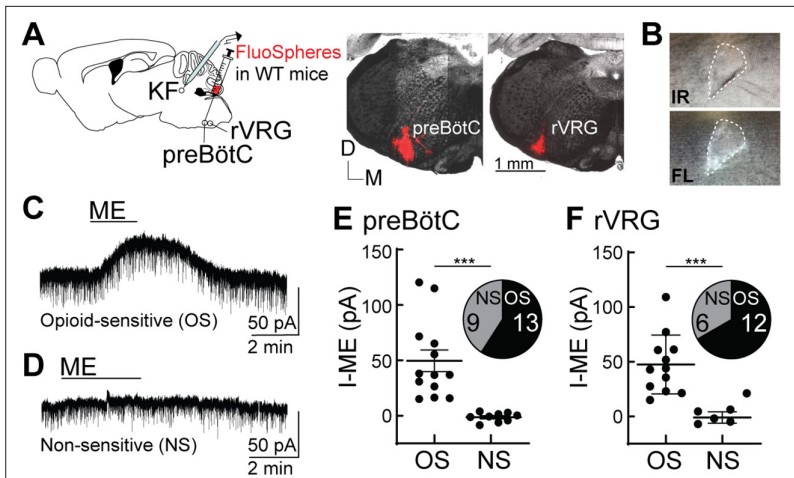

**Figure 4.** Opioids hyperpolarize Kölliker-Fuse (KF) neurons that project to the preBötzinger complex (preBötC) and rostral ventral respiratory group (rVRG). (**A**) Schematic (left) of approach to retrogradely label KF neurons that project to the preBötC or rVRG with FluoSpheres in wild-type mice. Images (right) of FluoSpheres in the injection area (preBötC or rVRG). The scale bar applies to both injection images. (**B**) A KF neuron retrogradely labeled by FluoSpheres shown with IR-Dodt and epifluorescent (FL) illumination. (**C, D**) Whole-cell voltage-clamp recordings from opioid-sensitive ('OS') and non-opioid-sensitive ('NS') retrogradely labeled KF neurons. Met-enkephalin (ME) (1 μM) induced an outward current in the opioid-sensitive (OS) neuron (**C**), but not the non-opioid-sensitive (NS) neuron (**D**). (**E, F**) Quantification of the amplitude of the ME-mediated current (I-ME [pA]) in OS and NS KF neurons that project to the preBötC (**E**; n = 22; ***p=0.0005; unpaired *t*-test) or the rVRG (**F**; n = 18; ***p=0.0007; unpaired *t*-test). ME induced an outward current in 13 of 22 KF neurons that project to the preBötC and 12 of 18 KF neurons that project to the rVRG. Individual data points are from individual neurons in separate slices. Line and error are mean ± SEM.

The online version of this article includes the following source data and figure supplement(s) for figure 4:

**Source data 1.** Opioid-mediated outward currents in Kölliker-Fuse (KF) neurons that project to the preBötzinger complex (preBötC) and rostral ventral respiratory group (rVRG).

**Figure supplement 1.** Opioids hyperpolarize Kölliker-Fuse (KF) neurons that project to the Bötzinger complex (BötC).

to excitatory neurons in the ventrolateral medulla express functional MORs on presynaptic terminals at a higher percentage than expected (91% [20 of 22 neurons]; *Figure 3*), we wanted to determine whether KF neurons also express functional somatodendritic MORs leading to hyperpolarization in a projection-specific manner. We recorded from KF neurons retrogradely labeled with FluoSpheres (580/605) that were unilaterally injected into the preBötC or rVRG of wild-type mice (*Figure 4A*). FluoSpheres were chosen over viral retrograde tracers for these experiments because they are highly visible in acute brain slices and do not spread as far in the injection area (*Figure 4A*), genetically alter neurons, require fluorescent amplification, or take long to express (2 d vs. 4 wk). Furthermore, Fluo-Spheres will label KF neurons regardless of *Oprm1* expression status, enabling us to determine the projection pattern of both *Oprm1*+ and *Oprm1*- neurons. Whole-cell voltage-clamp recordings were made from fluorescent KF neurons contained in acute brain slices (*Figure 4B*). The presence of an ME-mediated outward current identified KF neurons that express functional MORs and were opioid sensitive (OS) (*Figure 4C*) compared to neurons that lacked an ME-mediated outward current (non-sensitive [NS]) (*Figure 4D*). ME induced an outward current in 59% (13 of 22 neurons) of KF neurons that project to the preBötC (*Figure 4E*) and 67% (12 of 18 neurons) of KF neurons that project to the rVRG (*Figure 4F*). The average amplitude of the ME-mediated current was not different between KF neurons that project to preBötC (n = 13) or rVRG (n = 12) (p=0.8250; unpaired *t*-test). Thus, both opioid-sensitive and non-sensitive KF neurons project to preBötC and rVRG, with a proportion similar to the general population of KF neurons with unidentified projection targets (*Levitt et al., 2015*).

Given the potentially lesser degree of projections from *Oprm1*+ KF neurons to the BötC (*Figure 2— figure supplement 1*) and the ability to retrogradely label *Oprm1*-negative neurons with FluoSpheres, we also injected FluoSpheres into the BötC (n = 11) to test the hypothesis that *Oprm1*-negative KF

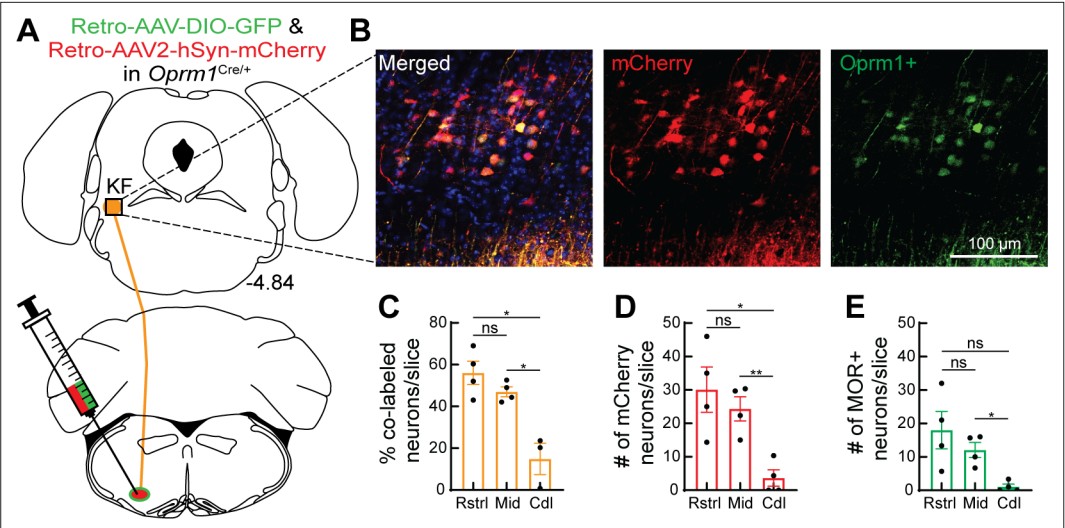

**Figure 5.** *Oprm1+* and *Oprm1-* dorsolateral pontine neurons project to the ventrolateral medulla. (**A**) Schematic of approach injecting retrograde virus encoding Cre-dependent GFP expression and a retrograde virus encoding mCherry expression into the ventrolateral medulla of *Oprm1*^Cre/+^ mice to label *Oprm1+* and *Oprm1-* dorsolateral pontine neurons that project to these respiratory nuclei. (**B**) Representative images of mCherry expression (retrogradely labels neurons regardless of *Oprm1* expression) and GFP expression (retrogradely labels *Oprm1+* neurons) in a rostral dorsolateral pontine slice (bregma –4.84 mm). (**C**) Summary of percentage of retrograde-labeled neurons that were *Oprm1+* (co-labeled with mCherry and GFP) in rostral (Rstrl, bregma –4.84 mm), mid-rostral (Mid, bregma –4.96 mm), and caudal (Cdl, bregma –5.2 mm) slices. (**D, E**) Summary of the average number of mCherry-expressing (**D**) or GFP-expressing MOR+ (**E**) dorsolateral pontine neurons per slice in rostral, mid-rostral, and caudal slices. Bar and error are mean ± SEM. Individual data points are from individual mice. N = 4 mice, three slices per region per mouse. *p<0.05, **p<0.01, ns = p>0.05 by one-way ANOVA and Tukey's post-test.

The online version of this article includes the following source data and figure supplement(s) for figure 5:

**Source data 1.** *Oprm1+* and *Oprm1-* dorsolateral pontine neurons project to the ventrolateral medulla.

**Figure supplement 1.** Medullary-projecting *Oprm1+* neurons are mostly absent from the caudal Kölliker-Fuse (KF) and lateral parabrachial areas.

neurons project to the BötC (*Figure 4—figure supplement 1*). We made whole-cell voltage-clamp recordings from fluorescent KF neurons and found that ME induced an outward current in only 36% (4 of 11 neurons) of KF neurons that project to the BötC (*Figure 4—figure supplement 1C*). Thus, a lower proportion of opioid-sensitive neurons project to BötC compared to preBötC and rVRG.

## Distribution of *Oprm1+* and *Oprm1-* dorsolateral pontine neurons projecting to the ventrolateral medulla

To further examine the distribution of *Oprm1+* and *Oprm1-* dorsolateral pontine neurons projecting to the ventrolateral medulla, retrograde AAV-hSyn-DIO-eGFP and retrograde AAV-hSyn-mCherry were unilaterally injected into the preBötC and rVRG of *Oprm1*^Cre/+^ mice (*Figure 5A*). Using this approach, projection neurons that express *Oprm1* will express GFP and mCherry, whereas projection neurons that do not express *Oprm1* will only express mCherry (*Figure 5B*). The number of mCherry and/or GFP-expressing neurons was evaluated in rostral (~bregma level –4.84 mm), mid-rostral (~bregma level –4.96 mm), and caudal (~bregma level –5.20 mm) sections of the dorsolateral pons (n = 4 mice, three slices per region per mouse). There were significantly more retrograde-labeled neurons in rostral and mid-rostral slices, regardless of *Oprm1* expression status (*Figure 5D*). Consistent with previous observations (*Figure 2*), retrograde-labeled *Oprm1+* neurons were mostly localized to the rostral and mid-rostral slices, and not in caudal slices or lateral parabrachial area (*Figure 5C and E* and *Figure 5—figure supplement 1*). The percentage of retrograde-labeled neurons that were *Oprm1+* (co-labeled with mCherry and GFP) in rostral slices (56%) and mid-rostral slices (47%) was higher than in caudal slices (15%) (*Figure 5C*). Taken together, *Oprm1+* and *Oprm1-* KF neurons that project to

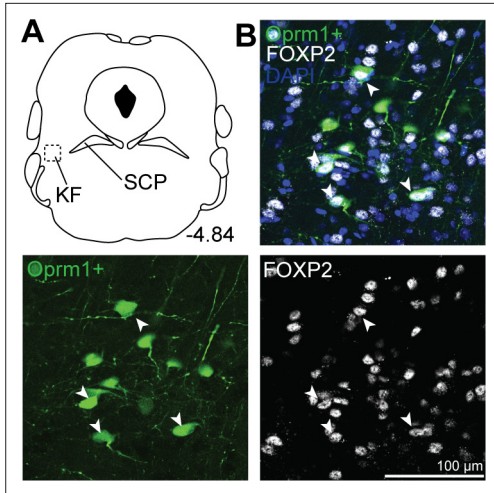

**Figure 6.** *Oprm1+*, medullary-projecting Kölliker-Fuse (KF) neurons express Forkhead box protein P2 (FoxP2). *Oprm1+* neurons that project to the ventrolateral medulla were retrogradely labeled by injection of retrograde AAV-DIO-GFP into *Oprm1*^Cre/+ mice. Immunohistochemistry was used to label FoxP2. (**A, B**) In rostral slices (bregma –4.84), FoxP2 is expressed in *Oprm1+* KF neurons that project to the ventrolateral medulla. Schematic (**A**) depicts the approximate bregma level and imaging area (dotted boxed area). The scale bar applies to all images. SCP, superior cerebellar peduncle.

The online version of this article includes the following figure supplement(s) for figure 6:

**Figure supplement 1.** Forkhead box protein P2 (FoxP2) expression in caudal Kölliker-Fuse (KF), but not external lateral parabrachial subnucleus.

respiratory nuclei in the ventrolateral medulla are distributed to the rostral and mid-rostral regions of the KF of mice.

## *Oprm1+*, medullary-projecting KF neurons express FoxP2, but not CGRP

Rostral glutamatergic KF neurons express FoxP2 (Forkhead box protein P2) (*Geerling et al., 2017*; *Karthik et al., 2022*), whereas MOR-expressing glutamatergic neurons in the external lateral parabrachial subnucleus that project to the forebrain express *Calca*, a gene that encodes the neuropeptide calcitonin gene-related peptide (CGRP) (*Huang et al., 2021*). Considering this, we performed immunohistochemistry for FoxP2 and CGRP on *Oprm1+* KF neurons projecting to the ventrolateral medulla. *Oprm1+*, medullary-projecting KF neurons expressed FoxP2 (n = 3; *Figure 6*), consistent with the population of glutamatergic FoxP2 and Lmx1b neurons in the rostral KF (*Karthik et al., 2022*). These are a separate population from FoxP2-expressing neurons located more dorsally and caudally in the inner portion of the external lateral parabrachial area and those activated by sodium deprivation (*Geerling et al., 2011*; *Karthik et al., 2022*). FoxP2 expression also overlapped with a smaller population of *Oprm1+* medullary-projecting neurons in the caudal KF, which contains GABAergic neurons (*Figure 6—figure supplement 1*; *Geerling et al., 2017*; *Karthik et al., 2022*). FoxP2 was not detected in the outer portion of the external lateral parabrachial subnucleus (*Figure 6—figure supplement 1*), consistent with previous findings (*Geerling et al., 2011*; *Karthik et al., 2022*).

*Oprm1+*, medullary-projecting KF neurons did not express CGRP (n = 3; *Figure 7*). Although CGRP expression was absent from the rostral KF and medullary-projecting *Oprm1+* neurons and neurites, it was robust in external lateral parabrachial neurons and their axon fiber projections (*Figure 7C and D*).

## Discussion

Opioid suppression of breathing could occur via multiple mechanisms and at multiple sites in the pontomedullary respiratory network. Here, we show that opioids inhibit an excitatory pontomedullary respiratory circuit via three mechanisms: (1) postsynaptic MOR-mediated hyperpolarization of KF neurons that project to the ventrolateral medulla, (2) presynaptic MOR-mediated inhibition of glutamate release from KF terminals onto excitatory preBötC and rVRG neurons, and (3) postsynaptic MOR-mediated hyperpolarization of the preBötC and rVRG neurons that receive pontine glutamatergic input (*Figure 8*). These mechanisms converge on a projection-specific opioid-sensitive circuit, whereby MOR-expressing excitatory KF neurons synapse onto MOR-expressing excitatory preBötC and rVRG neurons at a proportion that is higher than predicted based on MOR expression in either of these populations alone (*Bachmutsky et al., 2020*; *Kallurkar et al., 2022*; *Levitt et al., 2015*). We targeted the excitatory vglut2-expressing neurons in the ventrolateral medulla because they contain the populations of inspiratory rhythm-generating preBötC neurons (*Wallén-Mackenzie et al., 2006*; *Gray et al., 2010*; *Cui et al., 2016*) and inspiratory premotor rVRG neurons, and MOR deletion from

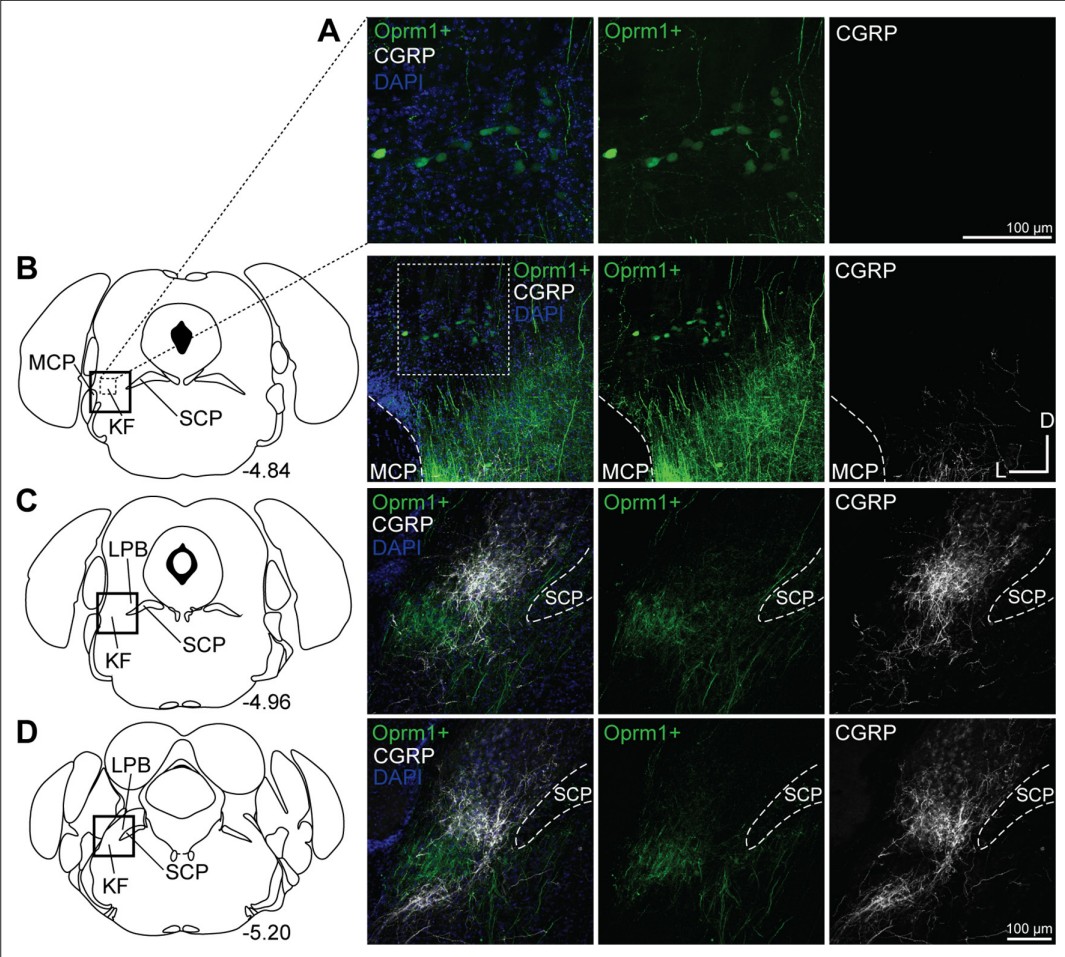

**Figure 7.** *Oprm1+*, medullary-projecting Kölliker-Fuse (KF) neurons do not express calcitonin gene-related peptide (CGRP). *Oprm1+* neurons that project to the ventrolateral medulla were retrogradely labeled by injection of retrograde AAV-DIO-GFP into *Oprm1*^Cre/+ mice. Immunohistochemistry was used to label CGRP. (**A, B**) CGRP is absent from rostral KF and *Oprm1+* KF neurons that project to the ventrolateral medulla (*Oprm1+*). (**C, D**) CGRP marks lateral parabrachial area (LPB) neurons and their axon fiber projections, but is absent from retrograde-labeled *Oprm1+* axon fiber projections in mid-rostral (**C**) and caudal (**D**) slices. The approximate bregma levels are to the right of each schematic. The images correspond to the dotted boxed area (row **A**) or the solid boxed area (rows **B–D**) of the slice schematic. The images in (**A**) are zoomed into the dotted boxed area of the image in (**B**). The scale bar in (**A**) applies to the images in row (**A**). The scale bar in (**D**) applies to images in rows (**B–D**). SCP, superior cerebellar peduncle; MCP, medial cerebellar peduncle.

vglut2 neurons prevents opioid-induced respiratory depression in medullary slices (*Sun et al., 2019*; *Bachmutsky et al., 2020*). Opioid inhibition of excitatory drive from KF onto these respiratory neuron populations is important for rhythm generation (preBötC) and respiratory pattern formation (rVRG). Thus, there are convergent mechanisms of opioid-induced respiratory suppression, including both presynaptic and postsynaptic opioid receptors in the dorsolateral pons and the ventrolateral medulla, resulting in distributed effects of opioids on the pontomedullary respiratory network.

## Opioid effects distributed throughout the pontomedullary respiratory network

The mechanistic insights shown here are parsimonious with previous studies examining the role of MORs in the dorsolateral pons (*Prkic et al., 2012*; *Levitt et al., 2015*; *Miller et al., 2017*; *Bachmutsky et al., 2020*; *Saunders and Levitt, 2020*; *Varga et al., 2020*; *Liu et al., 2021*) and the preBötC (*Gray et al., 1999*; *Sun et al., 2019*; *Bachmutsky et al., 2020*; *Varga et al., 2020*) in opioid-induced respiratory depression. Genetic deletion or pharmacological blockade of different subsets of pre- and

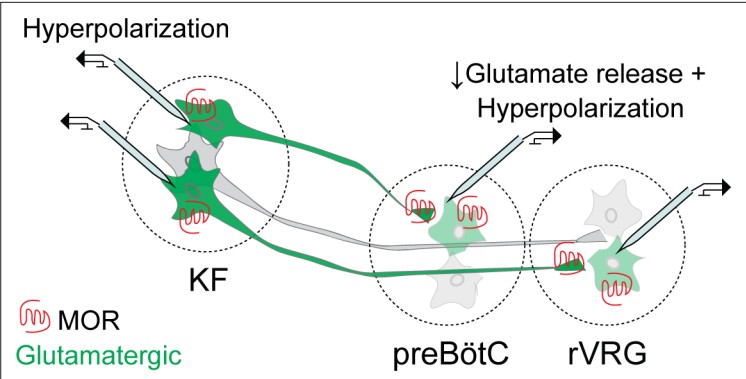

**Figure 8.** Summary schematic of mu opioid receptor (MOR) regulation of excitatory pontomedullary circuitry. Kölliker-Fuse (KF): Somatodendritic MORs hyperpolarize KF neurons that project to the ventrolateral medulla. Ventrolateral medulla: presynaptic MORs inhibit glutamate release from KF axon terminals onto glutamatergic preBötzinger complex (preBötC) and rostral ventral respiratory group (rVRG )neurons. Somatodendritic MORs hyperpolarize glutamatergic preBötC and rVRG neurons that receive KF input. Glutamatergic neurons are in green.

postsynaptic MORs in these areas mostly resulted in partial attenuation of opioid-induced respiratory rate suppression, presumably due to redundancy from the subset(s) of MORs in this pontomedullary circuit that were not deleted or blocked. Furthermore, additional MORs outside of the dorsolateral pontine and preBötC circuit likely contribute to respiratory suppression since deletion of MORs from both dorsolateral pons and preBötC did not eliminate morphine-induced respiratory suppression (*Bachmutsky et al., 2020*).

Often overlooked in the context of opioids, the rVRG contains abundant opioid receptors (*Lonergan et al., 2003*) and application of an opioid agonist into the rVRG suppresses rate and amplitude of phrenic nerve bursting (*Lonergan et al., 2003*; *Cinelli et al., 2020*). Here, we showed that MOR-expressing KF neurons densely project to the rVRG (*Figures 2 and 4*) and form glutamatergic synapses onto excitatory rVRG neurons (*Figure 3*). Presynaptic opioid receptors inhibit glutamate release from KF terminals synapsing onto rVRG neurons (*Figure 3*), and the excitatory rVRG neurons that receive glutamatergic input from the dorsolateral pons are hyperpolarized by postsynaptic opioid receptors (*Figure 3*). The impact of this highly opioid-sensitive projection on respiration warrants further investigation. Other respiratory-related areas in the medulla, such as the retrotrapezoid nucleus and the nucleus of the solitary tract (NTS), that receive *Oprm1*+ pontine input (*Liu et al., 2022*) could also be involved, but functional connectivity and impact remains to be determined. Another potential contributor to OIRD are the caudal medullary raphe nuclei since antagonism of opioid receptors in the dorsolateral pons, ventrolateral medulla, and caudal medullary raphe was able to eliminate remifentanil-induced respiratory depression (*Palkovic et al., 2022*).

## Opioids inhibit excitatory pontomedullary circuitry

Unexpectedly, KF neurons were more likely inhibited by presynaptic vs. somatodendritic MORs. The vast majority of KF terminals expressed presynaptic MORs since opioids inhibited glutamate release onto 91% of preBötC and rVRG neurons (*Figure 3*). In contrast, postsynaptic (somatodendritic) MOR-mediated outward currents were only observed in about two-thirds of medullary-projecting KF neurons (*Figure 4*), which matches prior studies without projection identification (*Levitt et al., 2015*; *Varga et al., 2020*). There are multiple possible reasons for this apparent heterogeneity. First, KF neurons may express MORs more abundantly on terminals than in the somatodendritic region. Second, KF neurons that did not have outward currents and were deemed not sensitive to opioids may express MORs, but lack GIRK channels, the functional readout we used to assess opioid sensitivity. MORs on these neurons could instead couple to other effectors, such as voltage-gated calcium channels (*Ramirez et al., 2021*). However, this seems unlikely since the percentage of retrograde-labeled neurons that were *Oprm1*+ (56% in rostral and 47% in mid-rostral slices; *Figure 5*) nearly matched the percentages of functionally identified opioid-sensitive KF neurons (59% of preBötC-projecting and 67% of rVRG-projecting neurons; *Figure 4*).

The last and most interesting possibility is that opioid-sensitive glutamatergic KF neurons preferentially synapse onto excitatory medullary neurons, while non-opioidergic KF neurons might preferentially synapse onto non-excitatory (i.e. inhibitory) medullary neurons. This hypothesis is consistent with anatomical-tracing studies showing that KF neurons project to excitatory and, to a lesser extent, inhibitory preBötC neurons (*Yang et al., 2020*), and could be tested by recording from labeled inhibitory neurons in the ventrolateral medulla. Inhibitory transmission in the medullary rhythm generator influences respiratory rate in the case of phasic inhibition or causes sustained apnea in the case of prolonged inhibition (*Baertsch et al., 2018*; *Cregg et al., 2017*; *Sherman et al., 2015*). We have recently found that inspiratory dorsolateral pontine neurons are silenced by fentanyl, whereas expiratory neurons are not (*Saunders et al., 2022*). An intriguing possibility is that opioid-insensitive pontine neurons, which have continued activity during opioid exposure, send prolonged input to inhibitory neurons in the ventrolateral medulla to promote apnea, perhaps using pathways overlapping those involved in apneas evoked by excitation of certain parts of the KF area (*Saunders and Levitt, 2020*; *Dutschmann and Dick, 2012*; *Dutschmann and Herbert, 2006*). This could include opioid-insensitive KF neurons that project to inhibitory neurons in the BötC since a higher proportion of opioid-insensitive pontine neurons projected to the BötC (*Figure 2* and *Figure 4—figure supplement 1*). Inhibitory input could also come from the NTS, which contains abundant MOR-expressing afferents and non-MOR-expressing neurons that are activated by disinhibition during opioid exposure (*Glatzer et al., 2007*; *Maletz et al., 2022*).

## Dorsolateral pontine subpopulations

The dorsolateral pons includes the lateral parabrachial area and the KF, both of which have been implicated in opioid-induced respiratory depression (*Levitt et al., 2015*; *Prkic et al., 2012*; *Varga et al., 2020*; *Liu et al., 2021*). Although effects of MORs in the lateral parabrachial and KF areas appear similar, mechanisms likely differ since the neuronal populations have different projection patterns (*Geerling et al., 2017*; *Huang et al., 2021*; *Liu et al., 2022*) and are involved in different behaviors besides breathing, especially the lateral parabrachial area, which has many different subpopulations (*Campos et al., 2018*; *Chen et al., 2018*; *Liu et al., 2022*; *Karthik et al., 2022*). In addition, the anatomical distinction between KF and lateral parabrachial area is not clear cut, though recent descriptions of transcription factor and neuropeptide/receptor expression in the dorsolateral pons provide opportunity to improve this (*Karthik et al., 2022*; *Pauli et al., 2022*).

The most well-defined area in the dorsolateral pons is the external lateral parabrachial subnucleus, which expresses Lmx1b and CGRP, but not FoxP2 (*Karthik et al., 2022*; *Huang et al., 2021*). CGRP-expressing external lateral parabrachial neurons project primarily to the forebrain (*Huang et al., 2021*) and are involved in pain processing, feeding, and $CO_2$-induced arousal (*Campos et al., 2018*; *Chen et al., 2018*; *Kaur et al., 2017*). Although MORs are highly co-expressed with CGRP in these neurons (*Huang et al., 2021*), we did not observe opioid-sensitive or *Oprm1*+retrograde-labeled neurons in the external lateral parabrachial area. We also did not observe a 'shell' pattern of retrograde-labeled *Oprm1*+ neurons surrounding the external lateral parabrachial area, in contrast with *Liu et al., 2022*, which could be due to slight differences in injection location, the fluorescent probe, and/or sensitivity of the experimental design. Rather, electrophysiologically or histologically identified opioid-sensitive/*Oprm1*+ neurons that project to the ventrolateral medulla were found rostrally and ventrally in the area overlapping FoxP2 expression in the KF. Thus, at least two distinct subpopulations of *Oprm1*+ dorsolateral pontine neurons exist that can be distinguished based on CGRP expression and projection pattern: forebrain-projecting CGRP-expressing neurons and medullary-projecting neurons that do not express CGRP. Both populations are involved in pain and breathing due, at least in part, to reciprocal excitatory synaptic connections (*Liu et al., 2022*). Although medullary-projecting *Oprm1*+ pontine neurons did not express CGRP (*Figure 7*), they can still be involved in pain processing, just not to the same extent as forebrain-projecting *Oprm1*/CGRP+ pontine neurons (*Liu et al., 2022*).

Both populations of *Oprm1*+ dorsolateral pontine neurons are also likely involved in opioid-induced respiratory depression. MORs in glutamatergic medullary-projecting rostral KF neurons could reduce respiratory rate by decreasing excitatory input to the preBötC and rVRG (*Figure 3*), leading to a distributed blunting effect on inspiration-generating processes within the ventrolateral medulla. In contrast, MORs in forebrain-projecting pontine neurons could reduce respiratory rate through intra-pontine excitatory connections with medullary-projecting MOR+ pontine neurons (*Liu et al., 2022*)

or through reduced excitatory input to forebrain areas involved in arousal (*Kaur et al., 2017*), which may be especially important in sleep-dependent effects of opioids on breathing (*Montandon and Horner, 2019*).

## PreBötC complex mechanisms

Significant attention has been given to the mechanisms of opioid suppression of inspiratory rhythm generation in the preBötC (*Sun et al., 2019*; *Bachmutsky et al., 2020*; *Baertsch et al., 2021*). Presynaptic opioid receptors in the preBötC inhibit synaptic transmission and have been postulated to disrupt preBötC neuron bursting (*Ballanyi et al., 2010*; *Wei and Ramirez, 2019*; *Baertsch et al., 2021*) by inhibition of excitatory neurotransmission that is dominant during bursts (*Ashhad and Feldman, 2020*), but the projection-specific location(s) of these presynaptic MORs is unknown. Our study has revealed a projection-specific presence of presynaptic MORs on glutamatergic terminals from dorsolateral pontine inputs to the preBötC. Although other MOR-expressing glutamatergic inputs are also likely contributors, including collaterals within the preBötC (*Rekling et al., 2000*), the role of these specific pontine inputs on opioid inhibition of respiratory rhythm generation is worthy of further investigation.

Only a subpopulation of preBötC neurons contain MORs (*Bachmutsky et al., 2020*; *Baertsch et al., 2021*; *Kallurkar et al., 2022*). The population of MOR-expressing preBötC neurons is heterogeneous, including nearly equal numbers of glutamatergic, GABAergic, and glycinergic neurons (*Bachmutsky et al., 2020*), type 1 and type 2 Dbx1-expressing inspiratory neurons (*Kallurkar et al., 2022*), and pre-inspiratory, inspiratory, expiratory, and tonic neurons (*Baertsch et al., 2021*). We found that MOR-expressing dorsolateral pontine glutamatergic inputs seem to preferentially synapse onto MOR-expressing excitatory preBötC neurons since 68% of preBötC neurons (8 of 12 neurons) that received glutamatergic input from the dorsolateral pons were hyperpolarized by opioid (*Figure 3G*). This percentage is higher than even the highest estimate of MOR-expressing preBötC neurons (*Baertsch et al., 2021*), suggesting dorsolateral pontine neurons preferentially target MOR-expressing glutamatergic preBötC neurons, which are important mediators of inspiratory rhythm generation and opioid-induced respiratory depression in medullary slices (*Sun et al., 2019*; *Bachmutsky et al., 2020*).

## Sensitivity and regulation of presynaptic and postsynaptic opioid receptors

Presynaptic and postsynaptic MORs couple to distinct effectors and are regulated differently, which can lead to differences in sensitivity that change with prolonged opioid exposure (*Coutens and Ingram, 2023*). For instance, postsynaptic, but not presynaptic, opioid receptors couple to GIRK channels (*Lüscher et al., 1997*) through binding of up to four Gβγ subunits directly to the channel (*Whorton and MacKinnon, 2013*). In contrast, presynaptic opioid receptors inhibit neurotransmitter release through inhibition of VGCCs (*Heinke et al., 2011*) or direct inhibition of vesicle release machinery (*Blackmer et al., 2001*; *Gerachshenko et al., 2005*). Coupling to these presynaptic effectors may be more sensitive since VGCCs can be inhibited by a single Gβγ subunit (*Zamponi and Snutch, 1998*) and vesicular release is steeply calcium dependent (*Katz and Miledi, 1967*). Consistent with this, presynaptic opioid receptor responses have higher sensitivity than postsynaptic responses when directly compared (*Pennock and Hentges, 2011*). Prolonged exposure to high doses of opioids can exacerbate differences in sensitivity since postsynaptic receptors desensitize more readily than presynaptic receptors (*Blanchet and Lüscher, 2002*; *Fyfe et al., 2010*; *Lowe and Bailey, 2015*; *Pennock et al., 2012*; *Rhim et al., 1993*). Thus, the responses of presynaptic receptors may predominate, especially after prolonged opioid exposure, for reasons related to receptor reserve, coupling to effectors and/or receptor regulation. The relative sensitivity of presynaptic and postsynaptic receptors in the pontomedullary circuit identified here will be important to determine, especially since postsynaptic opioid receptors on KF neurons are resistant to desensitization (*Levitt and Williams, 2018*), suggesting unique receptor regulation in these neurons.

In conclusion, our results show that opioids inhibit an excitatory pontomedullary respiratory circuit by three distinct mechanisms—somatodendritic MORs on dorsolateral pontine and ventrolateral medullary neurons and presynaptic MORs on glutamatergic dorsolateral pontine axon terminals in the ventrolateral medulla—all of which could influence distributed network function and contribute to the profound effects of opioids on breathing.

## Methods

### Animals

All experiments were approved by the Institutional Animal Care and Use Committee at the University of Florida (protocol #09515) and were in agreement with the National Institutes of Health 'Guide for the Care and Use of Laboratory Animals.' Homozygous *Oprm1*<sup>Cre/Cre</sup> mice (*Liu et al., 2021*) (Jackson Labs Stock #035574, obtained from Dr. Richard Palmiter, University of Washington) were crossed with homozygous Ai9-tdTomato Cre-reporter mice (*Rosa26*<sup>LSL-tdT/LSL-tdT</sup>) (Jackson Labs Stock #007909) to generate Oprm1-tdT mice. Homozygous vglut2-ires-Cre mice (Jackson Labs Stock #028863) were crossed with homozygous Ai9-tdTomato Cre-reporter mice (Jackson Labs Stock #007909) to generate vglut2-tdT mice. *Oprm1*<sup>Cre/+</sup>, Oprm1-tdT, vglut2-tdT, and wild-type C57BL/6J mice (male and female, 2–4 months old, weights commensurate with age and sex of normally developing C57BL/6J mice) were used for all experiments (*Table 1*). Mice were bred and maintained at the University of Florida animal facility. Mice were grouphoused with littermates in standard sized plastic cages and kept on a 12 hr light–dark cycle, with water and food available ad libitum.

### Stereotaxic injections

Mice (1–4 months old) were anesthetized with isoflurane (2–4% in 100% oxygen; Zoetis, Parsippany-Troy Hills, NJ) and placed in a stereotaxic alignment system (Kopf Instruments model 1900, Tujunga, CA). The dorsal skull was exposed and leveled horizontally in preparation for a small, unilateral craniotomy targeting either the KF (y = –5 mm, x = ±1.7 mm, z = - 3.9 mm from bregma), BötC (y = –6.6 mm and x = ±1.3 mm from bregma, z = - 5.625 mm), preBötC (y = –6.9 mm and x = ±1.3 mm from bregma, z = –5.625 mm), or rVRG (y = –7.2 mm and x = ±1.3 mm from bregma, z = –5.625 mm). Virus (undiluted) or FluoSpheres (580/605, 0.04 µm, diluted to 20% in saline, Invitrogen) were loaded into freshly pulled glass micropipettes and injected using a Nanoject III pressure injector (Drummond Scientific Company, Broomall, PA) at a rate of 10 nl every 20 s (100–200 nl total) (*Table 2*). Following the injection, the pipette was left in place for 10 min and slowly retracted. The wound was closed using Vetbond tissue adhesive (3M Animal Care Products, St Paul, MN). Mice received meloxicam (5 mg kg<sup>−1</sup> in saline, s.c.) and were placed in a warmed recovery chamber until they were ambulating normally. Mice were used either 2–6 d (FluoSpheres) or 4–5 wk (virus) later for electrophysiology, microscopy, or immunohistochemistry.

For retrograde labeling in *Oprm1*<sup>Cre/+</sup> mice, a 1:1 mixture (100 nl total) of either retrograde AAV-hSyn-DIO-eGFP (Addgene) and AAV2-hSyn-mCherry (UNC vector core) (*Figures 2*, *6*, and *7*, *Figure 2—figure supplement 1*) or retrograde AAV-hSyn-DIO-eGFP (Addgene) and retrograde AAV-hSyn-mCherry (Addgene) (*Figure 5*) was injected into the BötC, preBötC, and/or the rVRG. For labeling *Oprm1+* pontine neurons, AAV2-hSyn-DIO-EGFP (Addgene; 100 nl) (*Figure 1E–H*) was injected into the dorsolateral pons of *Oprm1*<sup>Cre/+</sup> mice. Vglut2-tdT mice received AAV2-hSyn-hChR2(H134R)-EYFP-WPRE-PA (Addgene; 100 nl) injections targeting the KF (*Figure 3*). Lastly, FluoSpheres (580/605, diameter: 0.04 µm, 20% in saline; 100 nl) were unilaterally injected into the BötC (*Figure 4—figure supplement 1*), preBötC or rVRG of wild-type C57BL/6J mice (*Figure 4*).

The correct placement of injections into the either the KF, BötC, preBötC, or rVRG was verified by anatomical landmarks, immunohistochemistry, and fluorescence in free-floating coronal brain slices (40–100 µm) using a MultiZoom microscope (Nikon AZ100). The BötC, preBötC ,and rVRG are located bilaterally in a rostro-caudal column in the ventrolateral medulla, just ventral to the nucleus ambiguous. The BötC, preBötC, and rVRG can be distinguished using the inferior olives, nucleus ambiguous, and nucleus tractus solitarius as medullary landmarks (*Franklin and Paxinos, 2008*; *Varga et al., 2020*). The KF is located bilaterally in the dorsolateral pons, just ventrolateral to the tip of the superior cerebellar peduncle and medial of the middle cerebellar peduncle (*Varga et al., 2020*; *Karthik et al., 2022*).

### Brain slice electrophysiology

Brain slice electrophysiology recordings were performed from KF neurons in acute brain slices from wild-type C57BL/6J mice (2–4 months old) or from vglut2-expressing preBötC and rVRG neurons in acute brain slices from vglut2-tdT mice (2–4 months old) injected with AAV2-hSyn-hChR2(H134R)-EYFP-WPRE-PA into the KF. Mice were anesthetized with isoflurane, decapitated, and the brain was removed and mounted in a vibratome chamber (VT 1200S, Leica Biosystems, Buffalo Grove, IL). Brain

slices (230 μm) containing either the KF, BötC, preBötC, or rVRG (identified based on anatomical landmarks and coordinates from *Franklin and Paxinos, 2008*) were prepared in warmed artificial cerebrospinal fluid (aCSF) that contained the following (in mM): 126 NaCl, 2.5 KCl, 1.2 $MgCl_2$, 2.4 $CaCl_2$, 1.2 $NaH_2PO_4$, 11 d-glucose, and 21.4 $NaHCO_3$ (equilibrated with 95% $O_2$–5% $CO_2$). Slices were stored at 32°C in glass vials with equilibrated aCSF. MK801 (10 μM) was added to the cutting and initial incubation solution (at least 30 min) to block NMDA receptor-mediated excitotoxicity. Brain slices were transferred to a recording chamber and perfused with 34°C aCSF (Warner Instruments, Hamden, CT) at a rate of 1.5–3 ml $min^{-1}$.

Cells were visualized using an upright microscope (Nikon FN1) equipped with custom-built LED-based IR-Dodt gradient contrast illumination and DAGE-MTI IR1000 camera. Cells containing Fluo-Spheres (580/605) or tdTomato were identified using LED epifluorescence illumination and a Texas Red filter cube (ex 559 nm/ em 630 nm). Whole-cell recordings were made using a Multiclamp 700B amplifier (Molecular Devices, Sunnyvale, CA) in voltage-clamp mode (Vhold = −60 mV). Glass recording pipettes (1.5–3 MΩ) were filled with internal solution that contained (in mM) 115 potassium methanesulfonate, 20 NaCl, 1.5 $MgCl_2$, 5 HEPES(K), 2 BAPTA, 1–2 Mg-ATP, 0.2 Na-GTP, adjusted to pH 7.35 and 275–285 mOsM. The liquid junction potential (10 mV) was not corrected. Data were low-pass filtered at 10 kHz and collected at 20 kHz with pCLAMP 10.7 (Molecular Devices), or collected at 400 Hz with PowerLab (LabChart version 5.4.2; AD Instruments, Colorado Springs, CO). Series resistance was monitored without compensation and remained <15 MΩ for inclusion. For optogenetic experiments, ChR2-expressing KF terminals were stimulated using 470 nm LED illumination (5 ms duration) through a ×40 objective to optogenetically evoke excitatory postsynaptic currents (oEPSC) in preBötC and rVRG neurons. A pair of optical stimuli (5 ms pulse, 50 ms interval) was delivered every 20 s. Blockers of glycine (strychnine, 1 μM) and GABA-A (picrotoxin, 100 μM) receptors were added to the aCSF to isolate excitatory neurotransmission. Peak amplitudes were determined in Clampfit 10.7 (Molecular Devices), and paired-pulse ratios (peak 2/peak 1), were determined in Microsoft Excel. All drugs were applied by bath perfusion. Bestatin (10 μM) and thiorphan (1 μM) were included with ME to prevent degradation.

## Immunohistochemistry and microscopy

Mice (2–4 months old) were anesthetized with isoflurane and transcardially perfused with phosphate-buffered saline (PBS) followed by 10% formalin. The brains were removed and stored at 4°C in cryo-protectant (30% sucrose in 10% formalin). A vibratome (VT 1200S, Leica Biosystems) was used to prepare free-floating coronal brain slices (40–100 μm) for microscopy or immunohistochemistry.

Free-floating slices were stained for forkhead box P2 (FoxP2), calcitonin gene-related peptide (CGRP), or neurokinin 1 receptor (NK1R) (*Table 3*). Slices were washed in diluting buffer (TBS with 2% bovine serum albumin, 0.4% Triton X-100, and 1% filtered normal goat serum) for 30 min, blocked in TBS and 20% normal donkey serum for 30 min, and incubated in primary antibody for 24 hr at 4°C. Primary antibodies included sheep polyclonal anti-FoxP2 (Cat# AF5647; R&D Systems, Minneapolis, MN; 1:1000 in diluting buffer), rabbit polyclonal anti-CGRP (Cat# T-4032; Peninsula, San Carlos, CA, 1:1000 in diluting buffer), and rabbit polyclonal anti-NK1R (Cat# S8305; Sigma-Aldrich; 1:1000 in diluting buffer). Slices were washed in diluting buffer and then incubated in secondary antibody (goat

**Table 1.** Mice used in this study.

| Strain | Reference | Source information | Key gene |
|---|---|---|---|
| *Oprm1*-cre | *Liu et al., 2021*. | Jax 035574<br>https://www.jax.org/strain/035574<br>Dr. Richard Palmiter (University of Washington) | Cre recombinase expressed in neurons with mu-opioid receptors |
| Vglut2-cre | *Vong et al., 2011* | Jax 028863<br>https://www.jax.org/strain/028863 | Cre recombinase expressed in excitatory glutamatergic neurons |
| Ai9, tdTomato Cre-reporter | *Madisen et al., 2010* | Jax 007909<br>https://www.jax.org/strain/007909 | LoxP-flanked STOP cassette preceding transcription of CAG promoter-driven red fluorescent protein variant (tdTomato) inserted into the Gt(ROSA)26Sor locus |
| C57BL/6J (wild-type) | *Simon et al., 2013* | Jax 000664<br>https://www.jax.org/strain/000664 | |

**Table 2.** Key resources.

| Injectate | Strain used | Injection target | Figure | Source Information |
|---|---|---|---|---|
| FluoSpheres 580/605, diameter: 0.04 µm | C57BL/6J | BötC, preBötC, or rVRG | *Figure 4* and *Figure 4—figure supplement 1* | Invitrogen |
| Retrograde AAV-hSyn-DIO-EGFP | *Oprm1*[Cre/+] | BötC, preBötC, or rVRG | *Figure 2* and *Figure 2—figure supplement 1* | Addgene |
| AAV2-hSyn-mCherry | *Oprm1*[Cre/+] | BötC, preBötC, or rVRG | *Figure 2* and *Figure 2—figure supplement 1* | UNC Vector Core |
| Retrograde AAV-hSyn-mCherry | *Oprm1*[Cre/+] | BötC, preBötC, and rVRG | *Figure 5* | Addgene |
| AAV2-hSyn-DIO-EGFP | *Oprm1*[Cre/+] | KF/PB | *Figure 1E–H* | Addgene |
| AAV2-hSyn-hChR2(H134R)-EYFP-WPRE-PA | vglut2-tdT | KF/PB | *Figure 3* | UNC Vector Core |

PreBötC, preBötzinger complex; rVRG, rostral ventral respiratory group; KF, Kölliker-Fuse; PB, parabrachial area.

**Table 3.** Antibodies used in this study.

| Antigen | Immunogen description | Source, host species, RRID | Concentration |
|---|---|---|---|
| Forkhead box P2 (FoxP2) | Targets human and mouse FoxP2 | R&D Systems, sheep polyclonal, Cat# AF5647, RRID:AB_2107133 | 1:1000 |
| Calcitonin gene-related peptide (CGRP) | Targets alpha-CGRP in canine, mouse, and rat | Peninsula, rabbit polyclonal, Cat# T-4032, RRID:AB_518147 | 1:1000 |
| Neurokinin 1 receptor (NK1R) | Targets C-terminal of NK1R in mouse, guinea pig, and human | Sigma-Aldrich, rabbit polyclonal, Cat# S8305 RRID:AB_261562 | 1:1000 |

anti-rabbit 647 [Cat# A32733; Thermo Fisher Scientific, Waltham, MA] or donkey anti-sheep 647 [Cat# A21448; Thermo Fisher Scientific; 1:500]) in diluting buffer. Finally, slices were rinsed with TBS and ddH$_2$0 and mounted onto glass slides with Fluoromount-G DAPI (Thermo Fisher Scientific). A confocal laser scanning microscope (Nikon A1R) with a ×10 objective (N.A. 0.3) or a multizoom microscope (Nikon AZ100) with a ×1 objective (N.A. 0.1) were used to image sections. All images were processed in Fiji (*Schindelin et al., 2012*).

To determine the spread and intensity of mCherry expression in the BötC, preBötC, and rVRG, serial coronal brain slices (50 µm) were collected and every slice containing mCherry expression was imaged in sequential order with a multizoom microscope (Nikon AZ100) at 500 ms exposure. Mean fluorescence intensity was determined for a region of interest drawn ventral to the NA to encompass the 7N/pFRG, BötC, preBötC, or rVRG in sequential slices. Mean intensity data were background subtracted and normalized to the peak intensity per injection. Bregma level was assigned using anatomical landmarks, including the inferior olives, nucleus ambiguus, and nucleus tractus solitarius (*Franklin and Paxinos, 2008*; *Varga et al., 2020*).

## Drugs

ME ([Met[5]]-enkephalin acetate salt), bestatin, DL-thiorphan, strychnine, picrotoxin, DNQX, 4-aminopyridine (4AP), and MK801 were from Sigma-Aldrich (St Louis, MO). Tetrodotoxin and ML-297 was from Tocris Bio-Techne (Minneapolis, MN). All drugs were applied by bath perfusion. Bestatin (10 µM) and thiorphan (1 µM) were included with ME to prevent degradation. ME is an endogenous opioid peptide agonist for mu and delta opioid receptors. Delta opioid receptors are not expressed in KF or preBötC neurons (*Varga et al., 2020*) and do not cause opioid-induced respiratory depression (*Dahan et al., 2001*). An EC80 concentration of ME (1–3 µM) was used to ensure robust and reliable responses but avoid acute receptor desensitization that occurs with higher concentrations (*Levitt and Williams, 2018*).

## Statistics

All statistical analyses were performed in GraphPad Prism 8 (La Jolla, CA). All error bars represent SEM unless otherwise stated. Replicates are biological replicates. Data with $n > 8$ were tested for normality with Kolmogorov–Smirnov tests. Comparisons between two groups were made using paired or unpaired two-tailed $t$-tests. Comparisons between three or more groups were made using one-way ANOVA followed by Tukey's post hoc test.

## Acknowledgements

This work was supported by the National Institutes of Health Grant R01DA047978 (ESL). JTB was supported by F31DA053798. We would like to thank Dr. Richard Palmiter (University of Washington) for generously providing the Oprm1-Cre mice and Keiko Arakawa for technical assistance. We thank Drs. Gordon Mitchell, John Williams, and Adrienn Varga for comments on the manuscript.

## Additional information

### Funding

| Funder | Grant reference number | Author |
| --- | --- | --- |
| National Institute on Drug Abuse | R01DA047978 | Erica S Levitt |
| National Institute on Drug Abuse | F31DA053798 | Jordan T Bateman |

The funders had no role in study design, data collection and interpretation, or the decision to submit the work for publication.

### Author contributions

Jordan T Bateman, Formal analysis, Funding acquisition, Investigation, Visualization, Writing - original draft, Writing – review and editing; Erica S Levitt, Conceptualization, Formal analysis, Supervision, Funding acquisition, Validation, Visualization, Writing – review and editing

### Author ORCIDs

Erica S Levitt http://orcid.org/0000-0002-3634-6594

### Ethics

All experiments were approved by the Institutional Animal Care and Use Committee at the University of Florida (protocol #09515) and were in agreement with the National Institutes of Health "Guide for the Care and Use of Laboratory Animals."

### Decision letter and Author response

Decision letter https://doi.org/10.7554/eLife.81119.sa1
Author response https://doi.org/10.7554/eLife.81119.sa2

## Additional files

### Supplementary files

• MDAR checklist

### Data availability

Data generated or analyzed during this study are included in the manuscript and supporting files.

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
