## [Editor Report]

Opioid-induced respiratory depression is one of the side effects of opioid drugs. Although opioid overdose deaths are highly prevalent, our knowledge of the neural circuits underlying respiratory depression in the brainstem is far from complete. The present study used a variety of sophisticated experimental techniques to convincingly reveal the identity of brainstem components that are part of the neural circuits involved in the mediation of opioid respiratory effects, together with defining potential synaptic underlying mechanisms. They focused on two regions of the brainstem, namely the Kolliker-Fuse and the preBötzinger Complex, and proposed a combination of three complementary processes at pre- and post-synaptic sites in both KF and preBötC regions to explain respiratory depression linked to opioid exposure. This study provides very important findings on the circuitry involved in opioid-induced respiratory depression, and the present results are of broad interest to the respiratory control research community, as well as medically relevant.

---

## [Decision Letter]

**Decision letter after peer review:**

Thank you for submitting your article "Opioid suppression of an excitatory pontomedullary respiratory circuit by convergent mechanisms" for consideration by *eLife*. Your article has been reviewed by 3 peer reviewers, and the evaluation has been overseen by a Reviewing Editor and Ronald Calabrese as the Senior Editor. The following individual involved in the review of your submission has agreed to reveal their identity: Gaspard Montandon (Reviewer #2). We strongly apologize for the abnormally long reviewing process. This was due in part to the difficulty to find reviewers available during the summertime and also due to personal problems that one reviewer got in the middle of the process, resulting in the need for us to give him some extra time allowing this reviewer to be able to provide his/her full review.

Essential revisions:

This study explores the mechanisms underlying inhibition by opioid drugs of two regions, the preBötzinger Complex, and the Kolliker-Fuse, well-known to mediate opioid-induced respiratory depression and respiratory rhythmogenesis. Based on sophisticated experiments three mechanisms are proposed to occur: postsynaptic inhibition of excitatory KF projection neurons, reduced glutamate release of KF projection neurons via MOR mediated pre-synaptic inhibition, and postsynaptic inhibition of medullary preBötC and VRG neurons. Despite the fact that conclusions are well supported by the data several important concerns must be addressed by the author:

1. Justify the dose (provide a dose-response curve) and the relevance (why not DAMGO) for having chosen ME at 3µM.

2. A detailed description of the recorded neuronal firing patterns should be provided in order to testify their belonging to the respiratory network.

3. Providing a summary diagram or a cartoon of the neural circuits between KF, preBötC, rVRG, and types of projections revealed here would greatly help following the demonstration and summarizing the present findings.

4. Orientate the discussion more towards broader network implications and not mainly focus on preBötC.

5. The possibility that other mechanisms may underlie inhibition by opioids that are not involving potassium current, such as inhibition of voltage-gated calcium channels should be considered.

6. The specificity and potential roles of the different sub-population of KF/LPB group should be more detailed.

7. The presentation (zoom) of the images of cells of interest should be improved.

8. Please provide a better description of the relationship between CGRP and FoxP2 and recorded neurons.

*Reviewer #1 (Recommendations for the authors):*

The 3 proposed mechanisms: postsynaptic inhibition of excitatory KF projection neurons, reduced glutamate release of KF projection neurons via MOR mediated pre-synaptic inhibition, and postsynaptic inhibition of medullary preBotC and VRG neurons are well supported by the data. However, most of the data presented were predictable and the postsynaptic mechanisms of MOR activation of KF and pre-BotC neurons were described before. That KF projection neurons are predominantly glutamatergic was also predictable from previous studies. Thus, it would have been of major interest to study whether the identified MOR-dependent pre- and post-synaptic mechanisms underlying opioid respiratory depression have different sensitivity to exogenous opioids.

The patch clamp recordings of either pontine or medullary neurons identify MOR effects via valid experimental protocols. However, only a single dose of 3uM ME was used. I think ME dose-response curves for all the patch clamp experiments would allow for a comparative analysis of the opioid sensitivity of post and pre-synaptic inhibition.

The ME dose response curves would increase the value of your study significantly and would help to straighten the discussion – at its present stage, the discussion is largely around in vitro mechanisms and specific subsets of pre-BotC neurons which is not much advancing the understanding of the network mechanisms of opioid respiratory depression. The discussion is hard to understand in the context of the designated roles of excitatory and inhibitory neurons in terms of concepts of rhythm generation and pattern formation. The essential roles of ponto-medullary synaptic interactions and distributed network mechanisms are largely avoided. Your work is clearly supporting that opioid respiratory depression is affecting distributed network functions and thus the extensive discussion concerned with the pre-BotC could be significantly shortened and the discussion then could be more broadly focused on the broader network implications.

Figure 1

Please check scale bars for C and E (-5.02). Also, the labelling pattern between C and E does not match – does Oprm1 Cre/tdT vs Oprm1 Cre/+ change the expression level? Please comment.

Figure 6

Foxp2 expression appears to be very high – the photographs indicate very high numbers of Foxp2 expression neurons in the KF that do not match previously published data. Did you validate the anti-body?

I suggest deleting the figure – at the end, it adds very little to the story.

*Reviewer #2 (Recommendations for the authors):*

This study explores the mechanisms underlying inhibition by opioid drugs of two regions, the preBötzinger Complex, and the Kolliker-Fuse, well-known to mediate opioid-induced respiratory depression. It shows pre-synaptic inhibition of Kolliker-Fuse glutamatergic neurons which projects to the preBötzinger Complex and rostral VRG. It also suggests that inhibition occurs at the preBötzinger Complex levels.

There are a few comments that the authors should address:

1) Pre-synaptic versus postsynaptic opioid effects. Presynaptic inhibition by ME is clearly demonstrated in this study. Postsynaptic inhibition is identified with outward currents. It is possible that other mechanisms may underlie inhibition by opioids that are not involving potassium current, such as the inhibition of voltage-gated calcium channels. The authors should acknowledge other potential mechanisms that may be involved.

2) Are the mechanisms described and identified here also mediating opioid-induced respiratory depression in-vitro (brainstem preparation) or in-vivo (anesthetized or freely-behaving animals)?

3) Activation of opioid receptors by MET5-enkephalin. Why did the authors use MET-enkephalin which is mostly acting on δ-opioid receptors with lower effects on mu-opioid receptors?

Why not use DAMGO to activate mu-opioid receptors or morphine to mimic opioid drug effects? How was the concentration chosen?

Do we know the affinity of ME on mu and δ receptors at this concentration?

4) FoxP2 and CGRP identify different types of KF/parabrachial neurons. What are the functional roles of these KF subpopulations?

If CGRP KF neurons are involved in co2-induced arousal and express opioid receptors, it may provide a better understanding of the effects of opioids on arousal (sedation) and the relationship between respiratory depression and sleep-wake states or sedation. This could be elaborated.

5) I would suggest the authors prepare a summary diagram or cartoon summarizing the neural circuits between KF, preBötC, rVRG, types of projections, etc… This would provide a clearer picture of the circuitry.

6) The size of microscopy images could be substantially increased for clarity and visibility. Zoom-in images could also help see cell bodies etc. Please see the comments below.

Page 2, line 5. The projection target and synaptic connections are (?) unknown.

Figure 1. Larger images of oprm1 expression could be provided on the right side of the figures, so individual cells could be easily visualized.

Figure 2A. The diagram could be simplified and could show only the projections identified in the study.

Figure 2B. Is it coexpressed with NK1-R? A larger version of NK1-R could be provided to better see cells.

Figure 3A. Separate images with NK1R and oprm1 could be provided. It is difficult to see the different expressions with images with 3 colors.

Page 5. Figure 3E-G. The results describing Figure 3E-G are not consistent with the legend of Figure 3. The effect of ME should be presented before Figure 3K etc. This section needs to be reorganized for clarity and consistency.

Figure 3E. Please define ME in legend.

Figure 4. Why is the concentration of ME lower in these experiments?

Figure 7. The absence of co-expression of oprm1 and cgrp is difficult to visualize with the current figures.

Page 9. Line 24. Considering the MOR inhibits neuronal activity through other mechanisms than GIRK channels, it is possible that MOR inhibition is due to calcium channel inhibition in some neurons.

[Editors' note: further revisions were suggested prior to acceptance, as described below.]

Thank you for resubmitting your work entitled "Opioid suppression of an excitatory pontomedullary respiratory circuit by convergent mechanisms" for further consideration by *eLife*. Your revised article has been evaluated by Timothy Behrens (Senior Editor) and a Reviewing Editor.

The manuscript has been improved but there are some remaining issues that need to be addressed, as outlined below:

*Reviewer #1 (Recommendations for the authors):*

Overall the manuscript appears to be somewhat improved.

General comment:

Unfortunately, the authors did not address my main suggestions to test whether the different neuronal populations in pons and medulla or the different mechanisms linked to somatic or pre-synaptic MOR may have different sensitivities. The argument that lower doses of morphine showed inconsistencies indicates that there might have been more to explore.

Specific comments:

Introduction:

"With the prevalence of opioid overdose on the rise (Wilson et al., 2020; Mattson et al., 2021), understanding the mechanisms of opioid-induced respiratory depression is of particular importance to aid in development of countermeasures and/or analgesics that do not affect breathing".

How does your study aid the development of countermeasures when systemic opioids act on all 3 mechanisms at the same time?

Methods:

Please provide the age and weight of the mice used for slice preparation and tracing experiments.

Results:

Figure 1 there appears to be two populations of MOR-expressing neurons – for the non-experts please indicate the location lateral parabrachial (external lateral nucleus) and Kolliker-Fuse nucleus in the photographs and sematic drawings.

Figure 2D-I the staining seems diffuse on the photographs (soma, axons/dendrites, and terminals?) – the figure legend states retrograde labeled Oprm1-expressing neurons. Please provide higher magnification pictures to illustrate labelled neurons.

Figure 3? NK1R labelling for the NA? please clarify, – I thought NK1R label pre-BotC neurons while ChAT would mark the NA.

Figure 6

The density of FoxP2 labeled cells indicated in the seems to be very high. Compared to previous reports. While the transcription factor FoxP2 labels cell nuclei it seems that the soma size of Oprm1+ labelled neurons appear in a very similar range l? Please clarify.

Discussion

Paragraph Opioids inhibit excitatory pontomedullary circuitry

"The last and most interesting possibility is that opioid-sensitive glutamatergic KF neurons preferentially synapse onto excitatory medullary neurons, while opioid-non-sensitive KF neurons are GABAergic and/or synapse onto non-excitatory (i.e. inhibitory) medullary neurons".

As I understand you patched MOR-positive cells in the rostral mid-rostral KF while the literature suggests that GABAergic KF neurons are located in the caudal KF so what is the basis of your speculation/hypothesis? I think this part of the discussion is not supported by data from the present study nor by the literature and should be at least toned down.

"An intriguing possibility is that opioid insensitive pontine neurons, which have continued activity during opioid exposure, send prolonged inhibitory input to the ventrolateral medulla to promote apnea".

Any evidence in the literature to support this speculation? What are the mechanisms for apnea mediated via descending inhibitory projections? Since there are only small fractions of inhibitory neurons in the caudal KF I would be cautious to suggest that these can mediate apnea via shifting excitatory-inhibitory balance in the pre-BotC microcircuit.

Para Dorsolateral pons: You should be firm with a statement that you have no evidence to support the findings of Lui et al., 2021.

---

## [Author Response]

Essential revisions:This study explores the mechanisms underlying inhibition by opioid drugs of two regions, the preBötzinger Complex, and the Kolliker-Fuse, well-known to mediate opioid-induced respiratory depression and respiratory rhythmogenesis. Based on sophisticated experiments three mechanisms are proposed to occur: postsynaptic inhibition of excitatory KF projection neurons, reduced glutamate release of KF projection neurons via MOR mediated pre-synaptic inhibition, and postsynaptic inhibition of medullary preBötC and VRG neurons. Despite the fact that conclusions are well supported by the data several important concerns must be addressed by the author:1. Justify the dose (provide a dose-response curve) and the relevance (why not DAMGO) for having chosen ME at 3µM.

Relevance: ME is the endogenous opioid peptide agonist for mu and delta opioid receptors. ME and DAMGO are both full agonists for mu opioid receptors. When we have compared ME and DAMGO using dose-response curves in KF neurons they were equally effective with EC50 values in the expected range, similar to neurons in the LC, periaqueductal gray and ventral tegmental area (Levitt et al., J Physiol, 2015). The main difference between DAMGO and ME is selectivity. DAMGO is selective for mu opioid receptors, while ME has equal potency for mu and delta opioid receptors. Delta opioid receptors are not expressed on KF neurons or preBötC neurons, since when MORs were genetically deleted from these neurons the effects of ME at these concentrations were eliminated (Varga et al., J Physiol, 2020). Delta opioid receptor activation does not cause respiratory depression, since respiratory depression caused by morphine, which activates both mu and delta opioid receptors, is eliminated in MOR knockout mice in our hands and as previously reported (Dahan et al., 2001). Thus, the selectivity of DAMGO is not needed for the experiments in this study making ME and DAMGO functionally identical. We prefer to use ME for brain slice experiments for several reasons. (1) ME is an endogenous opioid ligand and activation of opioid receptors by the endogenous ligand sheds light on opioid receptor functions independent of the exogenous opioid use context. (2) From a technical standpoint ME is superior to morphine, DAMGO or fentanyl because it rapidly washes from the slice allowing multiple applications and analysis of the “wash” period in comparison to the baseline period to determine if technical artifacts have occurred during the course of the recording.

Concentration: The concentration of ME (3 µM) is approximately the EC80, based on our previous concentration-response experiments in KF neurons (Levitt et al., 2015). We chose this concentration because we knew from previous work that it would give a reliable response in both KF and preBötC neurons (Varga et al., 2020) and avoid causing acute desensitization which can occur with concentrations at and above 10 µM (Levitt and Williams, 2018). In pilot experiments we used lower concentrations of ME (0.3 – 1 µM) followed by 3 µM. The response to these lower concentrations was not as consistent. Since our goal for these experiments was to determine opioid-sensitive versus non-sensitive responses, we wanted to use a high enough concentration to ensure robust and reliable responses, but not too high to avoid receptor desensitization. Clarifying text was added to the Drugs section of the Methods.

2. A detailed description of the recorded neuronal firing patterns should be provided in order to testify their belonging to the respiratory network.

These recordings were performed in non-rhythmic slices, so the firing pattern of the recorded neurons does not provide the respiratory-related information required to determine the respiratory phenotype of each neuron. The viral injections used to achieve regional specificity necessitate using adult mice, which precluded the use of rhythmic slices. To our knowledge, medullary rhythmic slices for preBötC neuron recording are still only possible from early postnatal mice. Rhythmic slices for the pons are not possible, and en bloc preparations are technically not feasible to combine with the recording and optical stimulation techniques used here. It is precisely for this reason that we recorded from vglut2-expressing preBötC and rVRG neurons, as a surrogate method of cell-type identification. This is an important caveat and we have added additional clarification of this limitation to the Results describing these experiments (Page 5, line 30-31).

“Because we could not determine the respiratory-related firing pattern of the neurons we recorded from in this study, we chose to target vglut2-expressing neurons since (1) this contains the population of inspiratory rhythm-generating preBötC neurons (Wallén-Mackenzie et al., 2006; Gray et al., 2010; Cui et al., 2016) and inspiratory premotor rVRG neurons, (2) KF neurons project to excitatory, more so than inhibitory, preBötC neurons (Yang et al., 2020), and (3) deletion of MORs from vglut2 neurons eliminates opioid-induced depression of respiratory output in medullary slices (Sun et al., 2019; Bachmutsky et al., 2020).”

3. Providing a summary diagram or a cartoon of the neural circuits between KF, preBötC, rVRG, and types of projections revealed here would greatly help following the demonstration and summarizing the present findings.

We made a summary schematic highlighting the major findings (Figure 8).

4. Orientate the discussion more towards broader network implications and not mainly focus on preBötC.

We have reorganized the Discussion to lead with expanded sections on implications of pontomedullary circuitry and dorsolateral pontine subpopulations. The section on preBötC mechanisms has been shortened and moved to the end of the Discussion.

5. The possibility that other mechanisms may underlie inhibition by opioids that are not involving potassium current, such as inhibition of voltage-gated calcium channels should be considered.

Inhibition of voltage-gated calcium channels is almost certainly involved in the presynaptic inhibition of neurotransmitter release by opioids. It is possible that opioid inhibition of dendritic VGCCs could affect other processes, such as synaptic integration or back-propagation of action potentials, but these have not been described for KF neurons. The possible involvement of VGCCs has been added to the Introduction and the Discussion.

Intro: “MORs inhibit neurotransmission by hyperpolarizing neurons through activation of somatodendritic GIRK channels and/or inhibiting presynaptic neurotransmitter release through inhibition of voltage-gated calcium channels (Jiang and North, 1992; Chahl, 1996; Zamponi and Snutch, 1998; Al-Hasani and Bruchas, 2011).”

Discussion: “Second, KF neurons that did not have outward currents and were deemed not sensitive to opioids may express MORs, but lack GIRK channels, the functional readout we used to assess opioid sensitivity. MORs on these neurons could instead couple to other effectors, such as voltage-gated calcium channels (Ramirez JM *et al.*, 2021).”

6. The specificity and potential roles of the different sub-population of KF/LPB group should be more detailed.

We have added a section on “Dorsolateral pontine subpopulations” to the Discussion. New Text is copied below:

“The most well-defined area in the dorsolateral pons is the external lateral parabrachial subnucleus, which expresses Lmx1b and CGRP, but not FoxP2 (Karthik et al., 2022; Huang et al., 2021). CGRP-expressing external lateral parabrachial neurons project primarily to the forebrain (Huang *et al.*, 2021), and are involved in pain processing, feeding, and CO2-induced arousal (Campos et al., 2018; Chen et al., 2018; Kaur et al., 2017). Although MORs are highly co-expressed with CGRP in these neurons (Huang et al., 2021), we did not observe opioid-sensitive or Oprm1+ retrograde labeled neurons in the external lateral parabrachial area. Rather, electrophysiologically or histologically identified opioid-sensitive/Oprm1+ neurons that project to the ventrolateral medulla were found rostrally and ventrally in the area overlapping FoxP2 expression in the KF. Thus, two distinct subpopulations of Oprm1+ dorsolateral pontine neurons exist that can be distinguished based on CGRP expression and projection pattern: forebrain-projecting CGRP-expressing neurons and medullary-projecting neurons that do not express CGRP. Both populations are involved in pain and breathing due, at least in part, to reciprocal excitatory synaptic connections (Liu *et al.*, 2022). Although medullary projecting Oprm1+ pontine neurons do not express CGRP (Figure 7), they can still be involved in pain processing, just not to the same extent as forebrain projecting Oprm1/CGRP+ pontine neurons (Liu *et al.*, 2022).

Both populations of Oprm1+ dorsolateral pontine neurons are also likely involved in opioid-induced respiratory depression. MORs in glutamatergic, medullary-projecting, rostral KF neurons could reduce respiratory rate by decreasing excitatory input to the preBötC and rVRG (Figure 3) leading to a distributed blunting effect on inspiration generating processes within the ventrolateral medulla. In contrast, MORs in forebrain-projecting pontine neurons could reduce respiratory rate through intra-pontine excitatory connections with medullary-projecting MOR+ pontine neurons (Liu et al., 2022), or through reduced excitatory input to forebrain areas involved in arousal (Kaur et al., 2017), which may be especially important in sleep-dependent effects of opioids on breathing (Montandon and Horner, 2019).”

7. The presentation (zoom) of the images of cells of interest should be improved.

Images in Figures 1, 2, 3, 6-supplement 1, and 7 have been improved according to suggestions specific to each Figure (described below).

8. Please provide a better description of the relationship between CGRP and FoxP2 and recorded neurons.

Recordings were made from retrogradely labeled neurons located rostral and ventral to the CGRP population of neurons, and in a similar location to the rostral/ventral population of FoxP2 neurons. We did not fill and identify recorded neurons post-hoc, so cannot say with certainty that they expressed CGRP or FoxP2.

We have added these sentences to the Discussion (page 10, lines 18-23):

“Although MORs are highly co-expressed with CGRP in these neurons (Huang et al., 2021), we did not observe opioid-sensitive or MOR+ retrograde labeled neurons in the external lateral parabrachial area. Rather, electrophysiologically or histologically identified opioid-sensitive MOR+ neurons that project to the ventrolateral medulla were found rostrally and ventrally in the area overlapping FoxP2 expression in the KF.”

Reviewer #1 (Recommendations for the authors):The 3 proposed mechanisms: postsynaptic inhibition of excitatory KF projection neurons, reduced glutamate release of KF projection neurons via MOR mediated pre-synaptic inhibition, and postsynaptic inhibition of medullary preBotC and VRG neurons are well supported by the data. However, most of the data presented were predictable and the postsynaptic mechanisms of MOR activation of KF and pre-BotC neurons were described before. That KF projection neurons are predominantly glutamatergic was also predictable from previous studies. Thus, it would have been of major interest to study whether the identified MOR-dependent pre- and post-synaptic mechanisms underlying opioid respiratory depression have different sensitivity to exogenous opioids.The patch clamp recordings of either pontine or medullary neurons identify MOR effects via valid experimental protocols. However, only a single dose of 3uM ME was used. I think ME dose-response curves for all the patch clamp experiments would allow for a comparative analysis of the opioid sensitivity of post and pre-synaptic inhibition.

As described above (Essential revision #1), ME (3 µM) is approximately the EC80, based on our previous dose-response experiments in KF neurons. We chose this concentration because it would provide robust and reliable responses and avoid causing acute desensitization which can occur with concentrations at and above 10 µM. We agree that it would be beneficial to examine the difference in sensitivity of pre-synaptic and post-synaptic responses. However, we think it would be most valuable to examine this difference in sensitivity in the context of acute versus chronic opioid exposure, which is out of scope of the current study and is best examined in a future study dedicated to this goal.

The ME dose response curves would increase the value of your study significantly and would help to straighten the discussion – at its present stage, the discussion is largely around in vitro mechanisms and specific subsets of pre-BotC neurons which is not much advancing the understanding of the network mechanisms of opioid respiratory depression. The discussion is hard to understand in the context of the designated roles of excitatory and inhibitory neurons in terms of concepts of rhythm generation and pattern formation. The essential roles of ponto-medullary synaptic interactions and distributed network mechanisms are largely avoided. Your work is clearly supporting that opioid respiratory depression is affecting distributed network functions and thus the extensive discussion concerned with the pre-BotC could be significantly shortened and the discussion then could be more broadly focused on the broader network implications.

We have significantly revised and reoriented the Discussion towards distributed pontomedullary mechanisms and circuitry. We have added a discussion of excitatory/inhibitory balance and expanded discussion of potential sources of opioid influence, including BötC, NTS, rVRG and raphe. We added a section on dorsolateral pontine subpopulations, including roles in breathing and pain, to address other reviewer comments. We shortened the section on preBötC mechanisms and moved it to the end of the Discussion.

Figure 1Please check scale bars for C and E (-5.02). Also, the labelling pattern between C and E does not match – does Oprm1 Cre/tdT vs Oprm1 Cre/+ change the expression level? Please comment.

The bregma level (-5.02) for C and E (now G) is accurate according to the mouse brain atlas (Franklin and Paxinos, 2008). The scale bars are different, as indicated in the figure legend, because the image in E (now G) is more zoomed in than C.

The fluorescent labeling in C and E (now G) are different because the experimental design was different. In figures A-D, Oprm1-Cre mice were bred with Ai9 tdT-Cre reporter mice. In the resulting offspring (Oprm1 Cre/tdT), Cre drives tdT expression in *all* Oprm1-expressing neurons in the brain throughout development, leading to tdT expression in both cell bodies and axon terminals projecting to the dorsolateral pons shown in these images. In new figures E-H, an AAV encoding Cre-dependent expression of GFP was injected into the KF/parabrachial area in adult Oprm1-Cre/+ mice, leading to GFP expression only in the neurons in the injection area (KF/PB), and not in axon terminals projecting to the KF/PB from elsewhere in the brain. The cell bodies are more visible in E-H because they are not obscured by the dense innervation from Oprm1-expressing axons into the area.

Figure 6Foxp2 expression appears to be very high – the photographs indicate very high numbers of Foxp2 expression neurons in the KF that do not match previously published data. Did you validate the anti-body?

The expression of FoxP2 we observed closely matches the recent comprehensive work from Karthik et al., 2022. FoxP2 expression in rostral PB/KF (bregma -4.9) is quite abundant, similar to Figure 6. In contrast, FoxP2 is excluded from the external lateral parabrachial subnucleus, which contains cells that project to the forebrain and express CGRP and Lmx1b and is located more caudally (bregma -5.2). We also observe this exclusion of FoxP2 from the external lateral parabrachial subnucleus, even though FoxF2 is detected in the surrounding PB. We have added representative images in Figure 6 —figure supplement 1. These images also show FoxP2 expression in caudal KF, consistent with Karthik et al., 2022 and overlapping with a small population of medullary projecting Oprm1+ cells.

I suggest deleting the figure – at the end, it adds very little to the story.

We hope that the additional figure supplement and text in the Results and discussion related to dorsolateral pontine subpopulations adds enough to the story to justify keeping these data in the manuscript.

Additional text in Results:

“Oprm1+, medullary-projecting KF neurons expressed FoxP2 (n=3; Figure 6), consistent with the population of glutamatergic FoxP2 and Lmx1b neurons in the rostral KF (Karthik *et al.*, 2022). These are a separate population from FoxP2 expressing neurons located more dorsally and caudally in the inner portion of the external lateral parabrachial area and those activated by sodium deprivation (Geerling et al., 2011; Karthik et al., 2022). FoxP2 expression also overlapped with a smaller population of Oprm1+ medullary projecting neurons in the caudal KF containing GABAergic neurons (Figure 6 – Figures supplement 1) (Geerling et al., 2017; Karthik et al., 2022). FoxP2 was not detected in the outer portion of the external lateral parabrachial subnucleus (Figure 6 —figure supplement 1), consistent with previous findings (Geerling et al., 2011; Karthik et al., 2022).”

Reviewer #2 (Recommendations for the authors):This study explores the mechanisms underlying inhibition by opioid drugs of two regions, the preBötzinger Complex, and the Kolliker-Fuse, well-known to mediate opioid-induced respiratory depression. It shows pre-synaptic inhibition of Kolliker-Fuse glutamatergic neurons which projects to the preBötzinger Complex and rostral VRG. It also suggests that inhibition occurs at the preBötzinger Complex levels.There are a few comments that the authors should address:1) Pre-synaptic versus postsynaptic opioid effects. Presynaptic inhibition by ME is clearly demonstrated in this study. Postsynaptic inhibition is identified with outward currents. It is possible that other mechanisms may underlie inhibition by opioids that are not involving potassium current, such as the inhibition of voltage-gated calcium channels. The authors should acknowledge other potential mechanisms that may be involved.

See response to Essential revision #5.

2) Are the mechanisms described and identified here also mediating opioid-induced respiratory depression in-vitro (brainstem preparation) or in-vivo (anesthetized or freely-behaving animals)?

Given the density of the projections we identified here and the importance of these brainstem areas in OIRD, we predict that these mechanisms would mediate OIRD in vitro and in vivo. We hope this study lays the groundwork for future studies assessing the impact of these projections on respiratory activity.

3) Activation of opioid receptors by MET5-enkephalin. Why did the authors use MET-enkephalin which is mostly acting on delta-opioid receptors with lower effects on mu-opioid receptors?Why not use DAMGO to activate mu-opioid receptors or morphine to mimic opioid drug effects? How was the concentration chosen?Do we know the affinity of ME on mu and delta receptors at this concentration?

See response to Essential revision #1. In addition, morphine is also a delta opioid receptor agonist, with similar affinity for mu and delta receptors, so it would have similar selectivity considerations as ME. Morphine is also a partial agonist (Levitt and Williams, Mol Pharmacol, 2012), which makes it difficult to see small effects. GIRK currents in mouse KF neurons can be relatively small and we did not know what size of effect would occur pre-synaptically. Since the goal of these experiments was to identify MOR+ projections, we were looking for robust results that would differentiate opioid-sensitive vs. non-opioid-sensitive responses, so we chose to use a full agonist (ME) at an ~EC80 concentration.

4) FoxP2 and CGRP identify different types of KF/parabrachial neurons. What are the functional roles of these KF subpopulations?If CGRP KF neurons are involved in co2-induced arousal and express opioid receptors, it may provide a better understanding of the effects of opioids on arousal (sedation) and the relationship between respiratory depression and sleep-wake states or sedation. This could be elaborated.

We have added two paragraphs to the Discussion on dorsolateral pontine subpopulations, including functional roles. The new paragraphs are below:

“The most well-defined area in the dorsolateral pons is the external lateral parabrachial subnucleus, which expresses Lmx1b and CGRP, but not FoxP2 (Karthik et al., 2022; Huang et al., 2021). CGRP-expressing external lateral parabrachial neurons project primarily to the forebrain (Huang et al., 2021), and are involved in pain processing, feeding, and CO2-induced arousal (Campos, 2018; Chen, 2018; Kaur et al., 2017). Although MORs are highly co-expressed with CGRP in these neurons (Huang et al., 2021), we did not observe opioid-sensitive or Oprm1+ retrograde labeled neurons in the external lateral parabrachial area. Rather, electrophysiologically or histologically identified opioid-sensitive/Oprm1+ neurons that project to the ventrolateral medulla were found rostrally and ventrally in the area overlapping FoxP2 expression in the KF. Thus, two distinct subpopulations of Oprm1+ dorsolateral pontine neurons exist that can be distinguished based on CGRP expression and projection pattern: forebrain-projecting CGRP-expressing neurons and medullary-projecting neurons that do not express CGRP. Both populations are involved in pain and breathing due, at least in part, to reciprocal excitatory synaptic connections (Liu et al., 2022). Although medullary projecting Oprm1+ pontine neurons do not express CGRP (Figure 7), they can still be involved in pain processing, just not to the same extent as forebrain projecting Oprm1/CGRP+ pontine neurons (Liu et al., 2022).

Both populations of Oprm1+ dorsolateral pontine neurons are also likely involved in opioid-induced respiratory depression. MORs in glutamatergic medullary projecting rostral KF neurons could reduce respiratory rate by decreasing excitatory input to the preBötC and rVRG (Figure 3) leading to a distributed blunting effect on inspiration generating processes within the ventrolateral medulla. In contrast, MORs in forebrain-projecting pontine neurons could reduce respiratory rate through intra-pontine excitatory connections with medullary-projecting MOR+ pontine neurons (Liu et al., 2022), or through reduced excitatory input to forebrain areas involved in arousal (Kaur et al., 2017), which may be especially important in sleep-dependent effects of opioids on breathing (Montandon and Horner, 2019).”

5) I would suggest the authors prepare a summary diagram or cartoon summarizing the neural circuits between KF, preBötC, rVRG, types of projections, etc… This would provide a clearer picture of the circuitry.

See response to Essential revision #3. Hopefully this new summary schematic (Figure 8) provides a clear picture of the circuitry and MOR effects.

6) The size of microscopy images could be substantially increased for clarity and visibility. Zoom-in images could also help see cell bodies etc. Please see the comments below.

Images have been enlarged and images displaying cell bodies have been added, as described in comments below.

Page 2, line 5. The projection target and synaptic connections are (?) unknown.

Corrected.

Figure 1. Larger images of oprm1 expression could be provided on the right side of the figures, so individual cells could be easily visualized.

Unfortunately, the density of MOR/tdT positive axonal projections into the pons obscures the ability to visualize individual cells in the Oprm1 Cre/tdT mice, even with higher magnification. This is not an issue for the experiments using viral mediated labeling of oprm1-expressing neurons, since only neurons in the injection area express GFP. We have now added a column to this figure showing labeling of oprm1 expressing neurons using the viral approach at all bregma levels, so that individual neurons can be visualized.

Figure 2A. The diagram could be simplified and could show only the projections identified in the study.Figure 2B. Is it coexpressed with NK1-R? A larger version of NK1-R could be provided to better see cells.

The diagram in 2A has been simplified and the NK1-R image in 2B has been enlarged.

Figure 3A. Separate images with NK1R and oprm1 could be provided. It is difficult to see the different expressions with images with 3 colors.

Separate images have been provided.

Page 5. Figure 3E-G. The results describing Figure 3E-G are not consistent with the legend of Figure 3. The effect of ME should be presented before Figure 3K etc. This section needs to be reorganized for clarity and consistency.

Figure 3 was reorganized to improve clarity and consistency with the text. The results of TTX and DNQX experiments have been moved to a supplemental figure to reduce the density of the figure and highlight the major opioid-related findings.

Figure 3E. Please define ME in legend.

Met-enkephalin (ME) has been defined in the legend.

Figure 4. Why is the concentration of ME lower in these experiments?

These experiments were done first using a concentration that we knew from prior experiments (Varga et al., 2020) would evoke reliable GIRK currents in KF neurons. The concentration of ME was increased for the experiments recording from medullary neurons to a concentration that would ensure reliable and robust responses in these neurons based on previous experience (Varga et al., 2020).

Figure 7. The absence of co-expression of oprm1 and cgrp is difficult to visualize with the current figures.

Zoomed in images have been added to help with visualization. Note, there is no CGRP expression in the image once we zoom in on the Oprm1+ neurons*.*

Page 9. Line 24. Considering the MOR inhibits neuronal activity through other mechanisms than GIRK channels, it is possible that MOR inhibition is due to calcium channel inhibition in some neurons.

See response to Essential Revision # 5 above. Specifically, this point in the Discussion was revised as follows: “Second, KF neurons that did not have outward currents and were deemed not sensitive to opioids may express MORs, but lack GIRK channels, the functional readout we used to assess opioid sensitivity. MORs on these neurons could instead couple to other effectors, such as voltage-gated calcium channels (Ramirez JM *et al.*, 2021).”

[Editors' note: further revisions were suggested prior to acceptance, as described below.]

The manuscript has been improved but there are some remaining issues that need to be addressed, as outlined below:Reviewer #1 (Recommendations for the authors):Overall the manuscript appears to be somewhat improved.General comment:Unfortunately, the authors did not address my main suggestions to test whether the different neuronal populations in pons and medulla or the different mechanisms linked to somatic or pre-synaptic MOR may have different sensitivities. The argument that lower doses of morphine showed inconsistencies indicates that there might have been more to explore.

While there might be more to explore, these experiments are not necessary to justify our conclusions that opioids inhibit an excitatory pontomedullary respiratory circuit via three mechanisms that are newly defined in a projection-specific manner, which were “well supported by the data”. In addition, our ability to do these experiments in a timely manner is constrained by a lack of mice and personnel since we just moved institutions and are still getting the lab back to normal operations. We have added a paragraph to the Discussion (below) to address the potential for differences in sensitivity based on presynaptic and postsynaptic receptors at other synapses. We ask for understanding and lenience from the reviewer regarding the necessity of these experiments and look forward to exploring this further in the future.

“Sensitivity and regulation of presynaptic and postsynaptic opioid receptors

Presynaptic and postsynaptic mu opioid receptors couple to distinct effectors and are regulated differently, which can lead to differences in sensitivity that change with prolonged opioid exposure (Coutens and Ingram, 2023). For instance, postsynaptic, but not presynaptic, opioid receptors couple to GIRK channels (Luscher et al., 1997) through binding of up to four Gβγ subunits directly to the channel (Whorton et al., 2013). In contrast, presynaptic opioid receptors inhibit neurotransmitter release through inhibition of VGCCs (Heinke et al., 2011) or direct inhibition of vesicle release machinery (Blackmer et al., 2005; Gerachshenko et al., 2005). Coupling to these presynaptic effectors may be more sensitive, since VGCCs can be inhibited by a single Gβγ subunit (Zamponi and Snutch, 1998) and vesicular release is steeply calcium dependent (Dodge and Rahamimoff, 1967). Consistent with this, presynaptic opioid receptor responses have higher sensitivity than postsynaptic responses when directly compared (Pennock and Hentges, 2011). Prolonged exposure to high doses of opioids can exacerbate differences in sensitivity, since postsynaptic receptors desensitize more readily than presynaptic receptors (Blanchet and Luscher, 2002; Fyfe et al., 2010; Lowe and Bailey, 2015; Pennock et al., 2012; Rhim et al., 1993). Thus, the responses of presynaptic receptors may predominate, especially after prolonged opioid exposure, for reasons related to receptor reserve, coupling to effectors and/or receptor regulation. The relative sensitivity of presynaptic and postsynaptic receptors in the pontomedullary circuit identified here will be important to determine, especially since postsynaptic opioid receptors on KF neurons are resistant to desensitization (Levitt and Williams, 2018), suggesting unique receptor regulation in these neurons.”

We were also prompted by the reviewer’s comment to carefully check the language about sensitivity throughout the manuscript and changed “sensitivity” to “proportion” in some instances and added quantification of these proportions. For instance: (Results, page 6, line 9) “presynaptic opioid receptors inhibit glutamate release from KF terminals onto a majority of excitatory preBötC and rVRG neurons (91 % 20 of 22 neurons).” And (Discussion page 8, lines 17-18): “These mechanisms converge on a projection-specific opioid-sensitive circuit, whereby MOR-expressing KF neurons synapse onto MOR-expressing excitatory preBötC and rVRG neurons at a proportion that is higher than predicted based on MOR expression in either of these populations alone (Bachmutsky *et al.*, 2020; Kallurkar *et al.*, 2022; Levitt *et al.*, 2015).” We would like to thank the reviewer for emphasizing this comment and think the additional text has helped further “straighten the Discussion”.

Specific comments:Introduction:"With the prevalence of opioid overdose on the rise (Wilson et al., 2020; Mattson et al., 2021), understanding the mechanisms of opioid-induced respiratory depression is of particular importance to aid in development of countermeasures and/or analgesics that do not affect breathing".How does your study aid the development of countermeasures when systemic opioids act on all 3 mechanisms at the same time?

As Reviewer 1 noted in the original round of reviews “The study advances our knowledge of network mechanisms that mediate opioid respiratory depression and may provide interesting frameworks for the development of therapies to counteract or prevent opioid respiratory depression.” However, we understand the point that we did not explore countermeasures (or analgesia), so we have revised this sentence as follows: *"With the prevalence of opioid overdose on the rise (Wilson et al., 2020; Mattson et al.,2021), understanding the network mechanisms of opioid-induced respiratory depression is of particular importance".*

Methods:Please provide the age and weight of the mice used for slice preparation and tracing experiments.

Mice used for brain slice recordings and tracing experiments were 2-4 months old. This is in the Methods animals section and has been added to the beginning of the Methods sections on brain slice electrophysiology and immunohistochemistry and microscopy.

Mice used in this study are normally developing mice with weights commensurate with age and sex during normal development of C57BL/6J mice (20-30 g for 2–4 month-old mice). Since we did not routinely record body weight, we don’t feel comfortable stating a weight range in the Methods, and instead state “weights commensurate with age and sex of normally developing C57BL/6J mice”.

Results:Figure 1 there appears to be two populations of MOR-expressing neurons – for the non-experts please indicate the location lateral parabrachial (external lateral nucleus) and Kolliker-Fuse nucleus in the photographs and sematic drawings.

We are glad the reviewer noticed two populations of MOR-expressing neurons. We agree and have updated Figure 1 to indicate locations of the KF, LPB and PBel on the schematics and images.

Figure 2D-I the staining seems diffuse on the photographs (soma, axons/dendrites, and terminals?) – the figure legend states retrograde labeled Oprm1-expressing neurons. Please provide higher magnification pictures to illustrate labelled neurons.

We have added higher magnification pictures showing retrograde labelled neurons from injections into the preBotC, rVRG and BotC in Supplemental Figure 2-2. We also agree that the most intense area of GFP fluorescence is diffuse and contains a significant amount of cell processes. We have changed the figure legend to “Oprm1+ KF neurons and neurites retrogradely labeled from the preBotC and rVRG” to reflect this. We also refer to retrograde labeled Oprm1-expressing “neurons and neurites” later in the legend and in the main text of the manuscript.

Figure 3? NK1R labelling for the NA? please clarify, – I thought NK1R label pre-BotC neurons while ChAT would mark the NA.

NK1R expression is a commonly used a marker for preBotC, but NK1Rs are also found in the NA (as well as ChAT). To facilitate the review process, please see NK1R expression (message or protein) in NA and preBotC from the following sources: image in Allen Brain Atlas, Figure 2 from Montandon, Liu and Horner (Scientific Reports, 2016), Figure 5 from McKay and Feldman (Am J Resp Critical Care Med, 2007), Figure 1 from Gray et al., (Nature Neuroscience, 2001) and Figure 2 from Wang, Stornetta, Rosin and Guyenet (JCN, 2001). NK1R labeling was not the only marker we used for the NA (we also used cytoarchitecture), but the NK1R labeling was obvious in the slices that were immunostained for NK1R, which are in Figure 2B and Figure 3. Regarding Figure 3, we removed the panels in question since they do not contribute much and were a source of confusion.

Figure 6The density of FoxP2 labeled cells indicated in the seems to be very high. Compared to previous reports. While the transcription factor FoxP2 labels cell nuclei it seems that the soma size of Oprm1+ labelled neurons appear in a very similar range l? Please clarify.

We also observe that FoxP2 labels only the cell nuclei, and importantly does not fill the entirety of the Oprm1+, GFP labelled neurons. We have improved and enlarged the higher magnification images in Figure 6 to make this more apparent. The overlap of FoxP2 and DAPI is best observed in GFP negative neurons. The exclusion of FoxP2 from the cytosol of GFP labelled neurons is harder to visualize in many of the neurons because the nucleus takes up a large portion of the soma. The uppermost neuron indicated by an arrowhead has the largest soma in this image and is the easiest to see that FoxP2 does not fill the entire soma.

Because of the difficulty to qualitatively visualize the nuclear restriction of FoxP2, we also quantified the relative intensities of the DAPI, FoxP2 and GFP signals across labelled somas. Example profile plots for the indicated neuron (yellow line) is shown in Author response image 1. Notice that FoxP2 overlaps with the DAPI signal, marking the nucleus, and is excluded from the cytosolic portion of the soma (labeled with GFP).

**Author response image 1. sa2fig1:** Example intensity profile plots of the GFP, FoxP2 and DAPI signal across the neuron, indicated by the yellow line on the image.

In addition, we performed control experiments, where the primary antibody against FoxP2 was omitted, but all other conditions were identical and performed in parallel (Author response image 2). FoxP2 specific labeling was in neurons (labeled with Neurotrace) and lacking from slices where the primary antibody was omitted.

**Author response image 2. sa2fig2:** FoxP2 immunolabeling (magenta) in the KF area with (top row) and without (bottom row) the primary anti-FoxP2 antibody. Neurotrace (green) is a fluorescent Nissl stain and labels neurons.

I am wondering which previous reports the reviewer is referring to regarding the density of FoxP2 labeled cells. Our reading of the prior literature reports that FoxP2 expression is lesser in caudal PB areas, but the density increases in the dorsolateral pons in more rostral slices (see Karthik et al., 2022). The density of FoxP2 labeled cells we observe in the rostral KF area matches the Allen Brain Atlas and Karthik et al., JCN, 2022 at the rostral-caudal level we are investigating (bregma -4.84). To help facilitate the review process, we added low resolution widefield images (admittedly low quality) showing that FoxP2 staining is focally restricted, in a manner similar to the Allen Brain Atlas (Author response image 3).

**Author response image 3. sa2fig3:** FoxP2 immunolabeling in the rostral dorsolateral pons (~bregma -4.84). See also, ISH for *Foxp2* from the Allen Brain Atlas.

Also see pertinent Figure panels from Karthik et al., JCN, 2022 that show high density of FoxP2 expressing cells in the rostral KF (> 500 neurons at bregma -4.9).

DiscussionParagraph Opioids inhibit excitatory pontomedullary circuitry"The last and most interesting possibility is that opioid-sensitive glutamatergic KF neurons preferentially synapse onto excitatory medullary neurons, while opioid-non-sensitive KF neurons are GABAergic and/or synapse onto non-excitatory (i.e. inhibitory) medullary neurons".As I understand you patched MOR-positive cells in the rostral mid-rostral KF while the literature suggests that GABAergic KF neurons are located in the caudal KF so what is the basis of your speculation/hypothesis? I think this part of the discussion is not supported by data from the present study nor by the literature and should be at least toned down.

Good point regarding the GABAergic KF neurons. We have removed the possibility that non-opioidergic KF neurons are GABAergic, and revised the text as follows.

New text: “The last and most interesting possibility is that opioid-sensitive glutamatergic KF neurons preferentially synapse onto excitatory medullary neurons, while non-opioidergic KF neurons might preferentially synapse onto non-excitatory (i.e. inhibitory) medullary neurons. This hypothesis is consistent with anatomical tracing studies showing that KF neurons project to excitatory and, to a lesser extent, inhibitory preBötC neurons (Yang *et al.*, 2020), and could be tested by recording from labeled inhibitory neurons in the ventrolateral medulla.”

"An intriguing possibility is that opioid insensitive pontine neurons, which have continued activity during opioid exposure, send prolonged inhibitory input to the ventrolateral medulla to promote apnea".Any evidence in the literature to support this speculation? What are the mechanisms for apnea mediated via descending inhibitory projections? Since there are only small fractions of inhibitory neurons in the caudal KF I would be cautious to suggest that these can mediate apnea via shifting excitatory-inhibitory balance in the pre-BotC microcircuit.

We agree that direct inhibitory descending projections that promote apnea are unlikely. More likely, descending excitatory projections onto inhibitory medullary neurons could promote apnea, possibly through mechanisms overlapping those involved in post-inspiratory apneas observed by excitation of the KF. We have rephrased to indicate this possibility:

“An intriguing possibility is that opioid insensitive pontine neurons, which have continued activity during opioid exposure, send prolonged excitatory input to inhibitory neurons in the ventrolateral medulla to promote apnea, perhaps using pathways overlapping those involved in apneas evoked by excitation of certain parts of the KF area (Saunders and Levitt, 2020; Dutschmann and Dick, 2012; Dutschmann and Herbert, 2006). This could include opioid insensitive KF neurons that project to inhibitory neurons in the BötC, since a higher proportion of opioid insensitive pontine neurons projected to the BötC (Figure 2 and 4 supplements).”

We also removed the sentence regarding a shift in the excitatory-inhibitory balance. I think it may be possible that opioids could shift the excitatory-inhibitory balance if opioid-insensitive neurons are synapsing onto inhibitory neurons in the medulla (BotC or preBotC), but this is highly speculative and has been removed from the manuscript.

Para Dorsolateral pons: You should be firm with a statement that you have no evidence to support the findings of Lui et al., 2021.

I think the assertion that we have no evidence to support the findings of Lui *et al.* is a little extreme, since we were focusing on slightly different areas and many of the functional effects on breathing align with what we would have predicted from manipulating opioidergic neurons in the dorsolateral pons. The main difference between our study and theirs is that we don’t see labeling of the PBN “shell” with retrograde injections into the preBotC area. This could be due to differences in the injection location and/or the experimental design. Liu et al. injected rgAAV-EF1a-DIO-FLPo into the preBotC and AAV9-EF1a-fDIO-ChR2-eYFP into the PB. Because only a few copies of recombinase are needed to drive expression of flp, which will be amplified by fDIO-ChR2-YFP, this approach has higher sensitivity but with the potential for false positives (ie. if the retrograde AAV spreads outside of the intended injection area). In contrast, our approach (rgAAV-DIO-GFP) will have less sensitivity but with less potential for false positives and more potential for false negatives. It remains to be determined which is the case. In addition, the cellular distribution of YFP tagged ChR is different from soluble GFP, leading to differences in the fluorescence appearance in labeled neurons and dendrites. We have added a shortened version of these points to the Discussion: “We also did not observe a “shell” pattern of retrograde labeled Oprm1+ neurons surrounding the external lateral parabrachial area, in contrast with (Liu et al., 2022), which could be due to slight differences in injection location, the fluorescent probe and/or sensitivity of the experimental design.”